# MASSIVE ACTIVATIONS ARE THE KEY TO LOCAL DETAIL SYNTHESIS IN DIFFUSION TRANSFORMERS

**Chaofan Gan**[1,2]     **Zicheng Zhao**[1]     **Yuanpeng Tu**[3]     **Xi Chen**[3]     **Ziran Qin**[1]
**Tieyuan Chen**[1]     **Mehrtash Harandi**[2]     **Weiyao Lin**[1]*

[1]Shanghai Jiao Tong University, [2]Monash University, [3]The University of Hong Kong

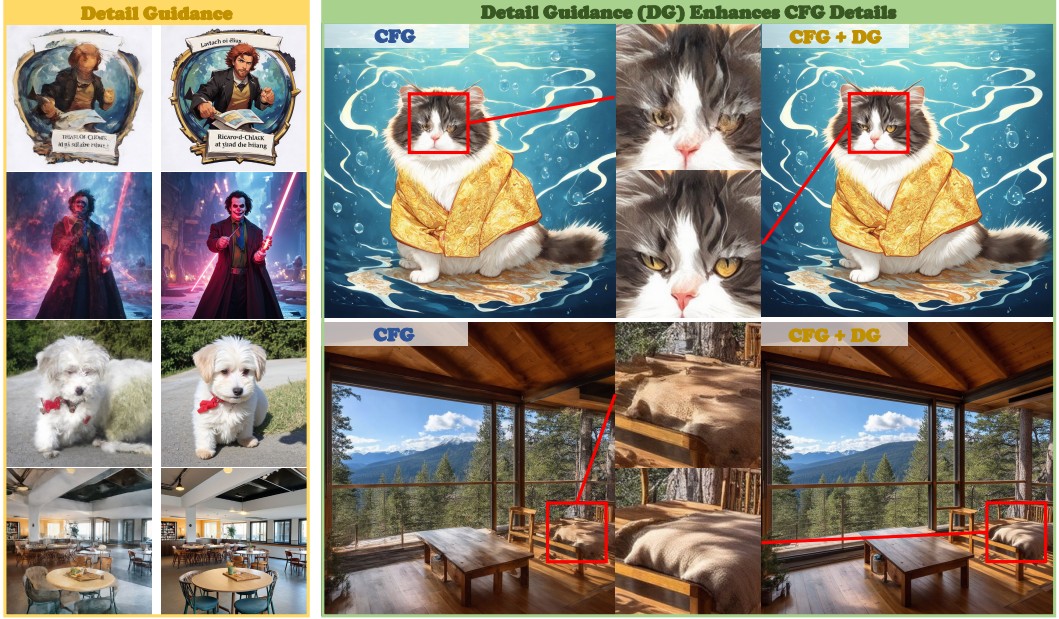

Figure 1: **Visual results of our Detail Guidance (DG).** Left: DG explicitly enhances fine-grained visual details, yielding high-quality outputs. Right: DG integrates seamlessly with Classifier-Free Guidance (CFG), allowing for further refinement of details.

## ABSTRACT

Massive Activations (MAs) are a well-documented phenomenon across Transformer architectures, and prior studies in both LLMs and ViTs have shown that they play a substantial role in shaping model behavior. However, the nature and function of MAs within Diffusion Transformers (DiTs) remain largely unexplored. In this work, we systematically investigate these activations to elucidate their role in visual generation. We found that these massive activations occur across all spatial tokens, and their distribution is modulated by the input timestep embeddings. Importantly, our investigations further demonstrate that these massive activations play a key role in local detail synthesis, while having minimal impact on the overall semantic content of output. Building on these insights, we propose **D**etail **G**uidance (**DG**), a MAs-driven, training-free self-guidance strategy to explicitly enhance local detail fidelity for DiTs. Specifically, DG constructs a degraded "detail-deficient" model by disrupting MAs and leverages it to guide the original network toward higher-quality detail synthesis. Our DG can seamlessly integrate with Classifier-Free Guidance (CFG), enabling joint enhancement of detail fidelity and prompt alignment. Extensive experiments demonstrate that our DG consistently improves local detail quality across various pre-trained DiTs (e.g., SD3, SD3.5, and Flux).

## 1 INTRODUCTION

Diffusion models (Rombach et al., 2022; Saharia et al., 2022) have recently achieved remarkable success across a wide range of generative tasks. Among various architectures, the Transformer (Vaswani

---

*Corresponding Author.                    **Project page:** https://ganchaofan0000.github.io/DG

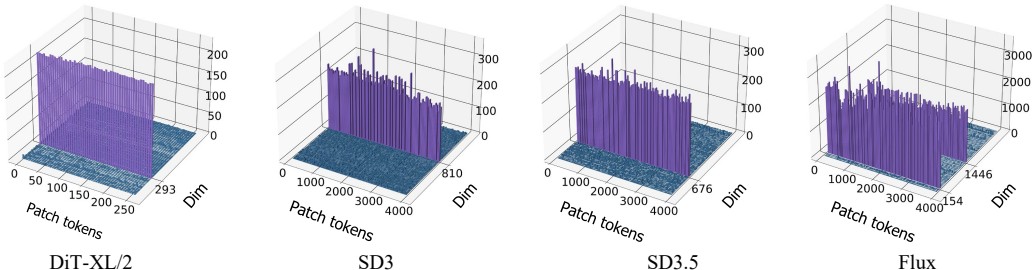

Figure 2: **Massive Activations in DiTs.** The activation magnitudes of internal hidden states from the middle block ($k = N/2$) and timestep ($t = T/2$). We present the average magnitudes over 1,000 text prompts. Massive Activations (MAs) are consistently concentrated in a few fixed dimensions across all image patch tokens. The MA dimensions remain consistent across all layers (see Figure 14).

et al., 2017) has emerged as a powerful and versatile backbone for diffusion models (Peebles & Xie, 2023), thanks to its flexibility and scalability. With the increasing availability of large-scale data and computational resources, many large Diffusion Transformers (DiTs) (Peebles & Xie, 2023; Esser et al., 2024) have recently emerged, achieving state-of-the-art performance in both image and video synthesis (Yang et al., 2024b; Hong et al., 2022; Wan et al., 2025).

Along with the rapid progress of DiTs, recent studies (Sun et al., 2024; Darcet et al., 2024; Gan et al., 2025) have uncovered an interesting phenomenon known as *Massive Activations* (MAs) in these Transformer-based models, where rare hidden activations exhibit unusually large magnitudes. Specifically, (Sun et al., 2024; Xiao et al., 2024) identifies the massive activations in Large Language Models (LLMs) and demonstrates that they are essential for long-context learning. Similar activation patterns are observed in Vision Transformers (ViTs), where they are utilized to process global semantic information (Darcet et al., 2024). More recently, several works (Gan et al., 2025; Fang et al., 2025) have reported the presence of massive activations in DiTs. However, their functional role within the visual generation process of DiTs remains largely unexplored.

In this paper, we aim to gain a deeper understanding of the role massive activations play in the visual generation tasks. We first conduct systematic investigations to study the characteristics of massive activations. Our investigations reveal that massive activations appear in a few fixed dimensions across all image tokens, which are text-independent ( Figures 2 and 4). In addition, we demonstrate that these activations are closely associated with the input timestep embeddings, where the timestep encoding can directly shape its distribution (Figure 4).

Furthermore, we perform activation intervention by disrupting the internal massive activations to directly investigate their impact on DiT generation. Our analysis shows that, when disrupting the massive activations, the visual output preserves consistent semantic content with the original images (Figure 5). These results suggest that the massive activations exert minimal influence on the semantics of the generation process. On the other hand, we found that the local details of visual output are significantly degraded when massive activations are disrupted, suggesting their crucial role in local detail synthesis (Figure 5). We propose the following interpretation to these findings: *DiT assigns massive activations to all spatial tokens to drive fine-grained local detail synthesis of each token, while timestep embeddings modulate these activations to adaptively control the detail synthesis process throughout generation.*

Motivated by these insights, we introduce **D**etail **G**uidance (**DG**), a MAs-driven, training-free self-guidance strategy for detail enhancement in DiT generation. Specifically, we construct a degraded "detail-deficient" network by disrupting the massive activations, and then leverage it to explicitly guide the original model toward generating higher-quality details. Our approach can be seamlessly integrated with classifier-free guidance (CFG), thereby achieving joint enhancement of detail fidelity and prompt alignment. Our main contributions can be summarized as follows.

- We provide a comprehensive study of massive activations in DiTs, demonstrating that these activations are crucial for fine-grained local detail synthesis while exerting minimal influence on the overall semantic content.
- We trace the massive activations to the influence of the input timestep embeddings, revealing that the input timestep encoding can directly shape their distribution.

- We introduce Detail Guidance (DG), a MA-driven, training-free self-guidance strategy to explicitly enhance local detail synthesis in DiTs. DG integrates seamlessly with Classifier-Free Guidance (CFG), leading to improved local detail synthesis.

## 2 RELATED WORK

### 2.1 DIFFUSION MODEL

Diffusion models (Rombach et al., 2022; Dhariwal & Nichol, 2021; Saharia et al., 2022) have become a dominant paradigm for high-quality visual synthesis. Early approaches primarily relied on U-Net architectures (Rombach et al., 2022) to model the denoising process. Recently, the field has shifted toward Transformer-based backbones (Vaswani et al., 2017). Among these advances, Diffusion Transformers (DiTs) (Peebles & Xie, 2023) have rapidly established themselves as a powerful backbone for visual generation. Due to the strong scalability and flexibility of transformer architecture, a new wave of large-scale DiTs(Esser et al., 2024; black-forest labs, 2024) (e.g., SD3, Flux) has emerged, achieving superior performance in various visual generation tasks (Yang et al., 2024b; Wan et al., 2025).

### 2.2 MASSIVE ACTIVATIONS

**Massive activations in LLMs.** Recent studies (Sun et al., 2024; Zhao et al., 2023; Xu et al., 2025) have identified the presence of massive activations in large language models (LLMs). These activations typically emerge at fixed dimensions of low-information tokens, such as starting or delimiter tokens. Importantly, some works (Xiao et al., 2024; Jin et al., 2025) have shown that massive activations contribute positively to contextual knowledge modeling, enabling LLMs to capture long-range dependencies more effectively. In addition, (Jin et al., 2025) traced the emergence of concentrated massive values into rotary position embeddings (RoPE).

**Massive activations in ViTs.** Similar activation patterns have also been observed in Vision Transformers (ViTs) (Darcet et al., 2024; Yang et al., 2024a; Sun et al., 2024). In ViTs, massive activations frequently arise in redundant background tokens and have been associated with encoding global semantic information. Moreover, (Yang et al., 2024a) traced the emergence of these activations to the influence of the input positional embeddings.

**Massive activations in DiTs.** Several studies on the acceleration of Diffusion Transformers (DiTs) (Liu & Zhang, 2024; Fang et al., 2025; Zhao et al., 2024) have highlighted the presence of outlier activations, whose extreme values pose a significant challenge for model quantization and distillation. More recently, DiTF (Gan et al., 2025) found that massive activations occur at fixed dimensions across all spatial tokens when employing DiTs as feature extractors, and showed that these activations substantially influence the discriminative quality of extracted features. However, the function of these massive activations in visual generation remains largely unexplored.

### 2.3 SAMPLING GUIDANCE FOR DIFFUSION MODELS.

Classifier-free guidance (CFG) (Ho & Salimans, 2022) has become the standard guidance mechanism for diffusion sampling. It extrapolates between the conditional and unconditional branches to amplify conditioning signals, thereby enhancing controllability and improving semantic alignment. Recently, a number of advanced strategies (Fan et al., 2025; Chung et al., 2024; Shen et al., 2024; Sadat et al., 2024; Zhang et al., 2024) have been proposed to further improve its effectiveness. For example, (Shen et al., 2024) introduces content-adaptive guidance strengths for different semantic components, while (Zhang et al., 2024) propose a frequency-aware guidance mechanism that adaptively modulates different frequency bands. Beyond CFG, auto-guidance (Karras et al., 2024) introduced a self-guidance signal by guiding the base model with a deliberately degraded "bad" version (e.g., reduced capacity or under-trained checkpoints), steering sampling toward higher-quality outputs. Other approaches (Ahn et al., 2024; Hong et al., 2023; Hyung et al., 2025) construct degraded predictions by perturbing internal mechanisms such as modifying attention maps or skipping blocks to bias the sampler toward a better image distribution.

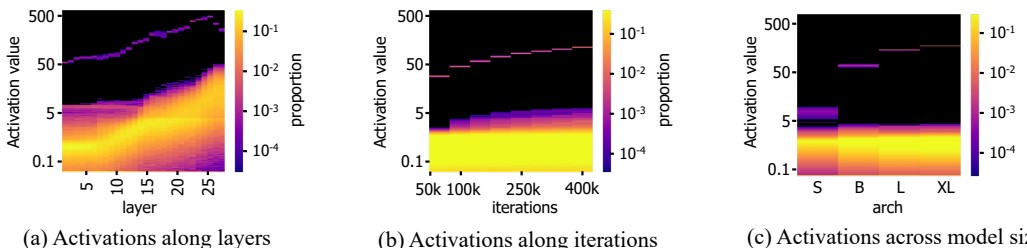

(a) Activations along layers  (b) Activations along iterations  (c) Activations across model size

Figure 3: **Illustration of several properties of massive activations in DiT-XL.** (a) Activation distribution of the hidden states along DiT layers (b) Activation distribution of the hidden states along training iterations (c) Activation distribution of the hidden states across different model sizes. Massive activations occur throughout all layers and persist across different model sizes.

## 3 PRELIMINARIES

**Diffusion models.** Diffusion models (Ho et al., 2020; Karras et al., 2022) generate data by progressively denoising Gaussian noise, starting from $z_T \sim \mathcal{N}(0, I)$. Given a clean sample $x \sim p_{\text{data}}(x)$, the forward diffusion process can be expressed as $z_t = x + \sigma(t)\,\epsilon$, where $\sigma(t)$ denotes the noise schedule and $\epsilon \sim \mathcal{N}(0, I)$. To learn the reverse process, a denoising network $D_\theta(z_t, t, c)$ is trained to predict the injected noise at each step, where $c$ is the conditioning signals (e.g., class labels or text prompts).

**Classifier-Free Guidance.** Classifier-Free Guidance (CFG) (Ho & Salimans, 2022) enhances diffusion model quality by jointly training the denoising network in conditional $D_\theta(z_t, t, c)$ and unconditional $D_\theta(z_t, t)$ modes. At sampling time, it combines the two predictions to amplify the conditioning signal:

$$\hat{D}_\theta(z_t, t, c) = D_\theta(z_t, t) + w\left(D_\theta(z_t, t, c) - D_\theta(z_t, t)\right) \tag{1}$$

where $w$ is the guidance scale. By extrapolating their difference, CFG strengthens semantic alignment and improves generation fidelity, but can sometimes lead to insufficient synthesis of fine-grained local details (Sadat et al., 2024; Chung et al., 2024).

**DiT architecture.** We provide the architecture of Diffusion Transformer (DiT) following (Peebles & Xie, 2023). For clarity, we omit the VAE component and focus on the latent diffusion transformer, denoted as $D_\theta = \{D_k\}_{k=1}^{N}$, where $k$ indexes the block and $N$ is the total number of blocks. Given noised latent $z_t \in \mathbb{R}^{C \times H \times W}$, the DiT block $D_k$ forward the internal hidden state $z_t^k$ through a residual connection (He et al., 2016) to the next block, formulated as

$$z_t^{k+1} = z_t^k + \alpha_k D_k(z_t^k, t, c) \tag{2}$$

where $\alpha_k \in \mathbb{R}^C$ denotes the dimension-wise residual scaling factor derived from the AdaLN layer (Perez et al., 2018). More architecture details can be found in Appendix A.

## 4 MASSIVE ACTIVATIONS IN DIFFUSION TRANSFORMERS

As shown in Figure 2, the hidden states of various DiTs consistently exhibit a prominent phenomenon: *Massive Activations* (MAs). This observation suggests that massive activations must play a crucial role in the visual generation process of DiTs. In this section, we conduct an in-depth investigation to understand *why* and *where* these massive activations emerge, and analyze their *role* in the visual generation process of DiTs.

### 4.1 CHARACTERISTICS OF MASSIVE ACTIVATIONS

**Massive activations occur throughout all layers across different model sizes.** We first investigate *where* massive activations emerge. As shown in Figure 3, we observed that massive activations exist in all internal DiT layers(Figure 3(a)). They emerge early during training (before 50k training iterations in Figure 3(b)), underscoring their crucial role in the internal computations of DiTs. Moreover, massive activations persist across models of different scales (Figure 3(c)). We present the layer properties of SD3, SD3.5, and Flux in Appendix B, which further confirm their presence throughout all internal blocks.

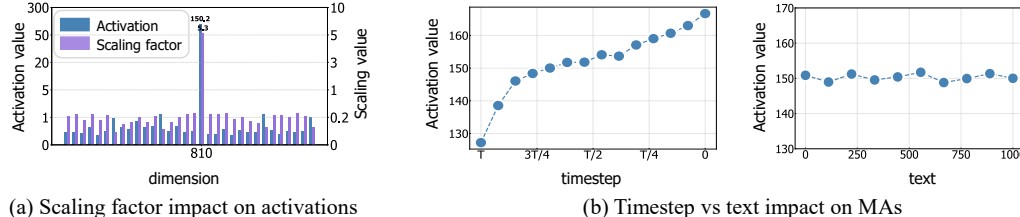

(a) Scaling factor impact on activations      (b) Timestep vs text impact on MAs

Figure 4: **Impact of the input timestep and text on Massive Activations (MAs) in SD3.** (a) Comparison of the distributions of hidden-state $z_t^k$ activations and their corresponding residual scaling factor $\alpha_k$. (b) Respective impact of input timestep and text embeddings on the magnitude distribution of MAs, where we compare the MAs of 1000 different text inputs. The massive activations are governed by the residual scaling factor; their magnitude is primarily shaped by the input timestep embedding $t$, while text embeddings $c$ have negligible effect.

**Massive activations appear in fixed dimensions across all patch tokens.** Then, we analyze the spatial distribution of massive activations, as illustrated in Figure 2. The results reveal that massive activations consistently appear at a fixed feature dimension (e.g., dimension 810 for SD3) across all spatial tokens. DiTF (Gan et al., 2025) has also characterized similar properties.

To delve into the massive activations in hidden states, we first examine the computation of hidden states $z_t^k \in \mathbb{R}^{C \times H \times W}$:

$$z_t^{k+1} = z_t^k + \alpha_k D_k(z_t^k, t, c), \alpha_k = \mathrm{MLP}_k(t, c) \tag{3}$$

where hidden states are computed via a residual connection, and $\alpha_k \in \mathbb{R}^C$ denotes the dimension-wise scaling factor parameters regressed by the AdaLN layer with an MLP network (see Appendix A for details). As shown in Figure 4(a), we compare the activation distributions of hidden states $z_t^k$ and the scaling factor $\alpha_k$ across dimensions. It can be observed that a prominent peak of $\alpha_k$ at dimension 810 leads to a corresponding concentration of massive activations (Figure 4(a)), indicating that the scaling factor $\alpha_k$ governs the dimension pattern of massive activations.

As the scaling factor $\alpha_k$ is produced by the AdaLN layer conditioned on the input timestep embedding $t$ and text embedding $c$ (see Equation (3)), we further examine how $t$ and $c$ respectively influence MAs (Figure 4(b)). This analysis leads to two key observations:

**Text embeddings have minimal impact on massive activations.** As shown in Figure 4(b), we compare the massive activation value across 1,000 different text prompts. We observe that these activations remain nearly constant (around 150) regardless of the input text embeddings, indicating that the text embeddings have negligible influence on the magnitude of the massive activations.

**Timestep embeddings shape the massive activations.** In contrast, we find that the timestep $t$ plays a dominant role for massive activation: the magnitude of massive activations increases steadily as $t$ decreases from $T$ to 0. We also get similar observations for SD3-5 and Flux (see Appendix C). These results suggest that massive activations in DiTs are mainly modulated by the timestep embeddings.

## 4.2 MASSIVE ACTIVATIONS FOR LOCAL DETAIL SYNTHESIS

Previous works (Darcet et al., 2024) have characterized massive activations in ViTs, showing that they primarily arise in specific tokens (e.g., background tokens) and serve to encode global information. In contrast, massive activations in DiTs occur across all spatial tokens. This fundamental difference naturally raises a key research question: *What role do massive activations play in DiTs?* To address this question, we perform activation intervention (Sun et al., 2024) to examine how massive activations influence the behavior of DiTs. Specifically, we manually disrupt the massive activation values in a single layer and then propagate the modified hidden state through the remaining DiT blocks. The results are presented in Figure 5. We provide the full activation intervention setting, including original, Non-MAs disrupted, and MAs-disrupted in Appendix D.

**Massive activations have minimal impact on semantic content.** We observe that the images generated by the MAs-disrupted model still preserve global semantics such as object identity, color composition, and overall layout, remaining consistent with those from the original model (Figure 5(a)). Moreover, the MAs-disrupted model maintains comparable prompt alignment metrics, achieving

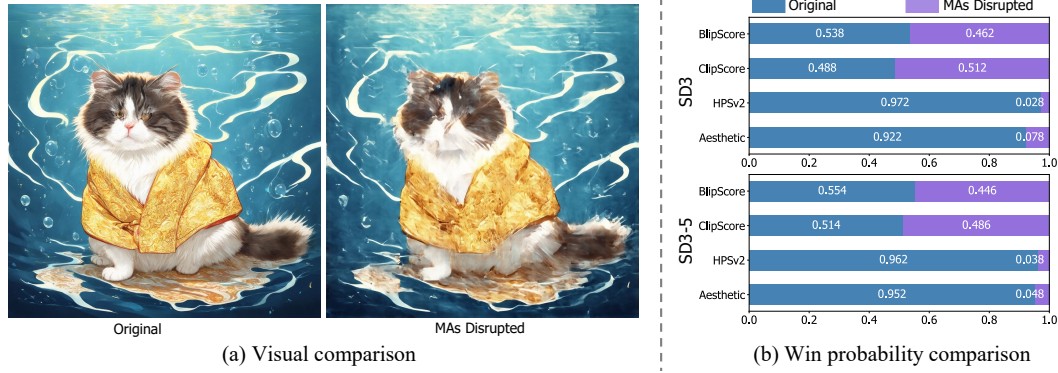

Original       (a) Visual comparison       MAs Disrupted

(b) Win probability comparison

Figure 5: **Comparison of the original and Massive Activations (MAs) disrupted models.** (a) Sampling results comparison between the original and MAs-disrupted models for SD3. (b) Win probability comparison for different models where we evaluate the model from two perspectives: **Prompt Alignment** (Blipscore and Clipscore) and **Local Detail Quality** (HPSv2.1 and Laion-Aesthetics). Disrupting massive activations markedly degrades the fidelity of fine-grained details in the generated images while exerting minimal impact on semantic content.

similar Blipscore (Li et al., 2022) (0.462 vs. 0.538) and Clipscore (Radford et al., 2021) (0.512 vs. 0.488) win probabilities relative to the original output (Figure 5(b)). These results indicate that the inherent massive activations exert minimal influence on the overall semantic content in the visual generation process of DiTs. This finding is consistent with the characteristic described in Section 4.1, where the input text embedding has negligible effect on massive activations.

**Massive activations play a key role in local detail synthesis.** More importantly, it can be easily observed that the fine-grained local details, including textures and subtle object parts (e.g., eyes and hair), are markedly degraded when massive activations are disrupted. Moreover, the MAs-disrupted model attains much lower win probabilities (0.028 on HPSv2.1 and 0.078 on Laion-Aesthetics) than the original model on the local detail quality metric, underscoring the crucial role of massive activations in fine-grained detail synthesis.

In combination with the characteristics of MAs described in Section 4.1, we summarize two key findings: (1) MAs are mainly shaped by the input timestep embedding, and (2) they are crucial for local detail synthesis. These findings are consistent with the generative dynamics of diffusion models (Ho et al., 2020; Rombach et al., 2022): the timestep embedding $t$ encodes the noise level and generation stage, with large $t$ guiding coarse structure reconstruction and small $t$ driving fine-grained refinement. As sampling proceeds from $T$ to 0, $t$ modulates the residual scaling factor $\alpha_k$, progressively amplifying massive activations (Figure 4(b)), which in turn orchestrate the detail synthesis process in DiTs.

## 4.3 DETAIL GUIDANCE FOR DIFFUSION TRANSFORMERS

Based on these findings, we make the following hypothesis: during training, DiT learns to assign massive activations to *all* spatial tokens to *drive* fine-grained local detail synthesis of each token, and uses timestep embeddings to *modulate* massive activations, thereby adaptively *orchestrating* the detail synthesis process throughout generation.

Motivated by these insights, we seek a *concise and effective* approach to exploit the capacity of MAs for enhancing fine-grained detail synthesis in DiTs. Accordingly, we propose a MAs-driven, training-free self-guidance strategy, termed Detail Guidance (DG). Our approach draws inspiration from the self-guidance mechanism (Karras et al., 2024), which guides the base model with a deliberately degraded "bad" version. Different from them, we construct the "bad" model by explicitly degrading its capacity for local detail synthesis.

Formally, let $D_\theta$ be the original pretrained DiT model and $z_t^k \in \mathbb{R}^{C \times H \times W}$ be the hidden state output of $k$-th block. We disrupt the massive activations in $z_t^k$ by masking (zeroing out) the corresponding dimensions to the massive activations and then propagate the modified hidden state $\tilde{z}_t^k$ through the remaining blocks. By disrupting the massive activations (drivers of local detail), we build a degraded

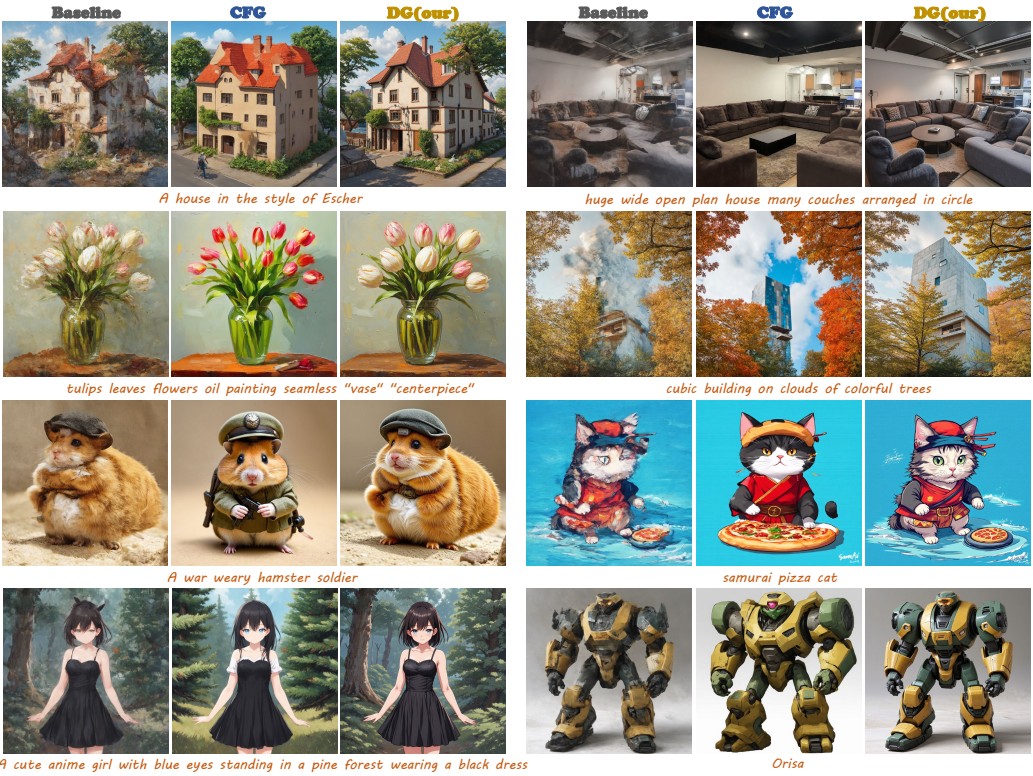

Figure 6: **Visual results of Detail Guidance (DG) on SD3.** Baseline denotes visual outputs without CFG. DG produces high-quality images with improved fine-grained details compared to Baseline. The CFG output is included as a reference for better comparison of detail quality.

model $D_{\theta,m}$ that produces detail-deficient outputs, where $m$ is the disrupted layer depth. Leveraging this degraded "detail-deficient" model $D_{\theta,m}$, we formulate our Detail Guidance (DG) following the diffusion self-guidance mechanism (Karras et al., 2024):

$$\hat{D}_\theta(z_t, t, c) = D_\theta(z_t, t, c) + w\big(D_\theta(z_t, t, c) - D_{\theta,m}(z_t, t, c)\big) \tag{4}$$

where $w$ controls the strength of the detail guidance. Our approach requires no extra training and can be directly applied to mostly pretrained DiT models ( Table 1).

**Integration with CFG.** Classifier-free guidance (CFG) (Ho & Salimans, 2022) is a standard technique that enhances semantic alignment by extrapolating between conditional and unconditional predictions. Our DG method is naturally complementary to CFG: whereas CFG strengthens semantic fidelity, DG explicitly enhances local detail quality. The combined guidance is expressed as

$$\hat{D}_\theta(z_t, t, c) = D_\theta(z_t, t, c) + \lambda\big(D_\theta(z_t, t, c) - D_\theta(z_t, t)\big)$$
$$+ w\big(D_\theta(z_t, t, c) - D_{\theta,m}(z_t, t, c)\big) \tag{5}$$

where $\lambda$ and $w$ are the guidance scales of CFG and DG, respectively.

## 5 EXPERIMENTS

### 5.1 EXPERIMENTAL SETUP

**Model Variants.** As our approach merely modifies internal hidden states, it can be readily applied to most pretrained DiTs without additional training or tuning. We evaluate DG on three representative text-to-image DiTs, SD3-Medium (Esser et al., 2024) (SD3), SD3.5-Large (Esser et al., 2024) (SD3.5), and Flux (black-forest labs, 2024). To comprehensively assess its effectiveness, we test DG under two settings: Conditional (Cond) generation without CFG and CFG generation. The default generated image size is 1024x1024. Further implementation details are provided in Appendix H.2.

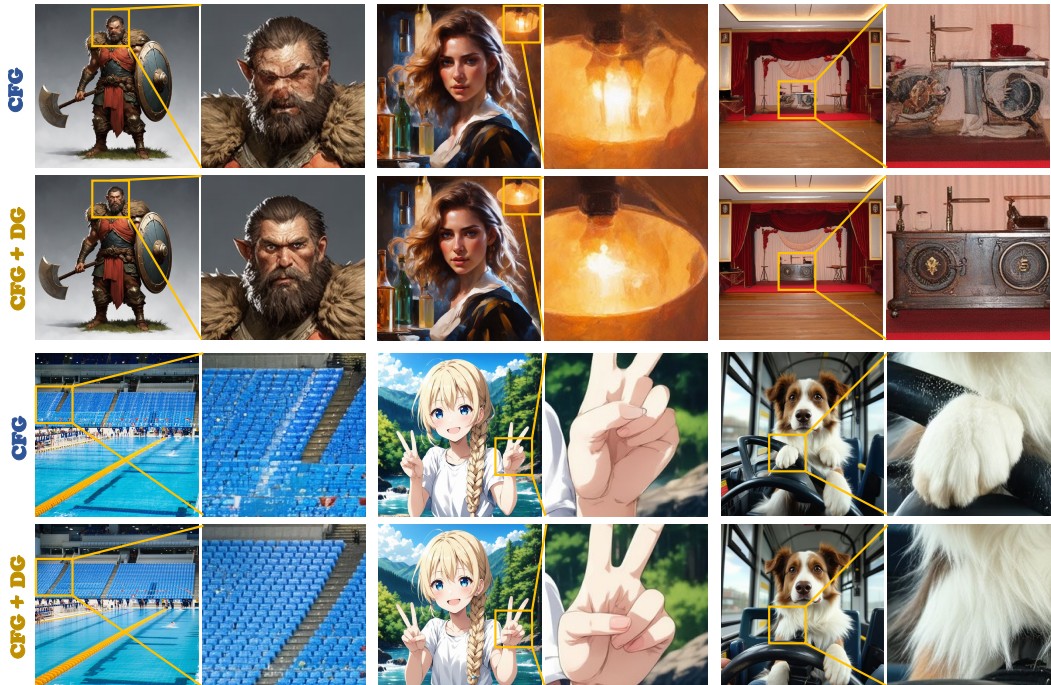

Figure 7: **Integration with CFG.** Rows 1-2: SD3; Rows 3-4: SD3.5. Incorporating DG into Classifier-Free Guidance (CFG) yields outputs with markedly richer and more refined local details.

| Model | Type | DG | Prompt Alignment | | Detail Quality | |
|-------|------|-----|-----------|-----------|---------|-----------|
| | | | Clipscore | Blipscore | HPSv2.1 | Aesthetic |
| SD3 | Cond | ✗ | 22.11 | 66.74 | 21.84 | 5.58 |
| | | ✓ | **24.15** | **76.52** | **28.65** | **6.01** |
| | CFG | ✗ | **26.64** | **87.01** | 28.23 | 5.80 |
| | | ✓ | 26.25 | 86.32 | **29.87** | **6.03** |
| SD3.5 | Cond | ✗ | 24.90 | 70.09 | 23.65 | 5.94 |
| | | ✓ | **26.01** | **83.66** | **29.23** | **6.16** |
| | CFG | ✗ | 27.67 | **92.62** | 29.9 | 6.01 |
| | | ✓ | **27.68** | 91.61 | **30.7** | **6.18** |
| Flux | Cond | ✗ | 22.09 | 57.60 | 19.33 | 5.50 |
| | | ✓ | **25.69** | **80.55** | **27.88** | **6.13** |
| | CFG | ✗ | 27.04 | **87.76** | 29.16 | 5.96 |
| | | ✓ | **27.14** | 86.23 | **29.25** | **6.12** |

Table 1: **Quantitative comparison on dataset Pick- a-Pic.** Our DG strategy brings substantial improvements on detail quality for both settings, demonstrating its effectiveness in enhancing visual details. The best highlights in bold.

**Datasets and Evaluation Metric.** We assess our method on two standard benchmarks: the Pick-a-Pic "test unique" split (Kirstain et al., 2023) and HPSv2.1 (Wu et al., 2023). To quantify prompt alignment, we compute Clipscore (Radford et al., 2021) and Blipscore (Li et al., 2022). To evaluate the fidelity of fine-grained local details, we adopt HPSv2.1 (Wu et al., 2023) and Laion-Aesthetics (Schuhmann, 2022) as quality metrics. Further evaluation details are provided in Appendix H.3.

## 5.2 MAIN RESULTS ON VISUAL GENERATION

**Evaluation of DG.** Table 1 reports the quantitative results of Detail Guidance (DG) on three pre-trained DiTs. DG achieves substantial improvements in both prompt alignment and detail quality, (e.g., Blipscore from 70.09 to 83.66 and Aesthetic from 5.94 to 6.16 on SD3.5). Qualitative results in Figure 6 further confirm that DG effectively enhances fine-grained local details while faithfully preserving the overall image structure. We provide qualitative results on SD3.5 and Flux in Appendix P.

| Train | Method | HPSv2.1 | | | | | Aesthetic |
| | | Anime | Concept | Painting | Photo | Avg. | |
|---|---|---|---|---|---|---|---|
| ✓ | CFG | 31.34 | 30.62 | 30.98 | 28.01 | 30.24 | 5.93 |
| | APG | 30.76 | 29.98 | 30.24 | 26.86 | 29.46 | 5.89 |
| | CFG++ | 31.58 | 30.32 | 30.95 | 27.24 | 30.02 | 5.91 |
| | Semantic-CFG | 30.92 | 29.99 | 30.92 | 29.16 | 30.25 | 5.89 |
| | FA-CFG | 31.07 | 30.10 | 31.09 | 28.76 | 30.26 | 5.96 |
| | CFG-Zero | 31.64 | 31.05 | **31.35** | 28.25 | 30.57 | 6.07 |
| ✗ | PAG | 30.59 | 28.92 | 29.38 | 27.91 | 29.20 | 6.10 |
| | DG (Ours) | 31.14 | 30.17 | 30.05 | 28.70 | 30.14 | **6.14** |
| ✓ | CFG+DG (Ours) | **32.23** | **31.11** | 31.27 | **29.21** | **30.96** | 6.13 |

Table 2: **Evaluation of different guidance on dataset HPSv2.1 with SD3.** Train means whether need to train an unconditional branch. The best highlights in bold, while the second best is underlined.

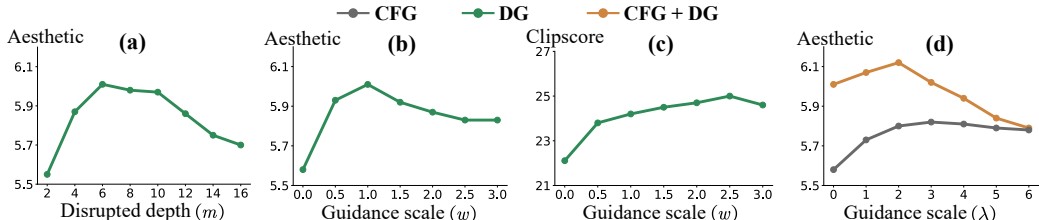

Figure 8: **Investigations of disrupted depth $m$, scales $\lambda$ and $w$ for SD3.**

Moreover, we report experiments on ImageNet 256×256 in Appendix I, showing that our DG strategy also enhances visual quality in class-conditional generation tasks.

**DG versus CFG.** From Table 1, DG yields higher detail-quality scores (e.g., Aesthetic 6.01 vs. 5.80 for SD3), whereas CFG achieves stronger prompt alignment. As illustrated in Figure 6, DG produces outputs with richer local textures, while CFG excels at semantic alignment. These results indicate that DG primarily enhances local detail synthesis, while CFG strengthens semantic alignment.

**Integrating DG with CFG.** DG integrates seamlessly with CFG, consistently improving detail quality as shown in Table 1. Visual comparisons in Figure 7 highlight that the combined strategy further refines fine-grained details, yielding higher overall image quality. We provide more visual results of SD3 and SD3.5 in Appendix O.

**DG versus PAG.** PAG generates the bad version by modifying the self-attention maps of the conditional branch and does not require training an unconditional branch. As shown in Table 2, DG achieves better performance than PAG (e.g., 30.14 vs. 29.20 on HPSv2.1 and 6.14 vs. 6.10 on Aesthetic), demonstrating its stronger effectiveness in enhancing visual details.

**DG versus advanced CFG.** We further compare DG with advanced CFG strategies such as FA-CFG, Semantic-CFG, and CFG-Zero (see Table 2). DG requires no unconditional-branch training, yet still obtains the highest Aesthetic score (6.14) and competitive HPSv2.1 performance among all methods. When combined with CFG, DG achieves the best results on both HPSv2.1 and Aesthetic metrics, underscoring its effectiveness and generality in improving visual quality.

### 5.3 ABLATION STUDY

**Disrupted depth $m$.** We examine the effect of the disrupted depth $m$ of massive activations, as shown in Figure 8(a). Our DG strategy achieves the best performance when applied to intermediate blocks (e.g., $m$ ranging from 4 to 10). We hypothesize that early blocks mainly contain heavy noise and lack even coarse image structures, making disruption there uninformative, while applying disruption in late blocks occurs too close to the final output and thus has minor impact on generation. Based on these observations, we primarily perturb massive activations in the intermediate layers and set the default $m = 6$ for the SD3 model. The configurations for SD3.5 and Flux are provided in Appendix H.2.

**Guidance scales $w$ and $\lambda$.** We present the quantitative results across different scales in Figure 8. Our DG consistently achieves stable and high Aesthetic (AES) scores (Figure 8(b)) and CLIPScore

| Baseline | Local DG | DG |
| --- | --- | --- |

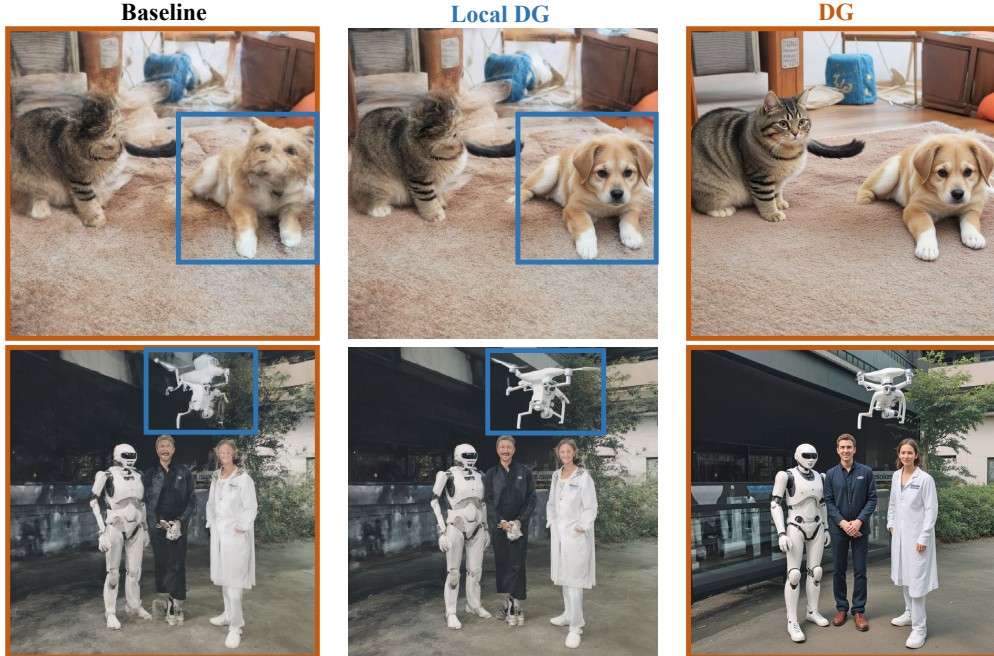

Figure 9: **Visual results of our Local DG on SD3.** We implement the Local DG strategy by selectively disrupting the MAs in tokens within a target region (e.g., the dog). Local DG significantly enhances the details of the local region (e.g., dog in the first row) while leaving other regions, such as the cat and background, essentially unaffected.

(Figure 8(c)). When combined with CFG, it further boosts AES performance (Figure 8(d)). These results highlight the effectiveness and robustness of our approach in enhancing fine-grained details.

**User study.** We conduct a user study to evaluate the benefits of our DG strategy from three key aspects: prompt alignment, color consistency, and detail preservation(see Appendix J). The results show that our DG strategy yields substantial improvements in color consistency and detail preservation, demonstrating its effectiveness in enhancing fine-grained visual details.

### 5.4 TOKEN-SPECIFIC GUIDANCE USING LOCAL DG

To further investigate the locality property of MAs, we design a Local DG strategy in which we mask MAs exclusively for a selected subset of spatial tokens, rather than masking MAs across all spatial tokens as in the original DG setting. As shown in Figure 9, Local DG effectively guides local details in the generated images. Specifically, by disrupting the Massive Activations only in the tokens corresponding to the dog, Local DG successfully enhances the local details of the dog while leaving other regions, such as the cat and background, essentially unaffected. These results clearly demonstrate the locality of massive activations: each MA primarily drives the local detail synthesis of its corresponding spatial token. By exploiting this property, we can achieve token-specific guidance using DG by selectively disrupting MAs in the target regions.

## 6 CONCLUSION

In this paper, we systematically investigate an intriguing phenomenon in DiTs, termed *Massive Activations* (MAs). We find that these activations emerge across all spatial tokens and that their distribution is shaped by the input timestep embeddings. Our further analysis demonstrates that these activations are critical for local detail synthesis in DiTs. We interpret them as drivers of local detail information whose magnitude is dynamically modulated by timestep embeddings, thereby orchestrating detail synthesis during the DiT generation process. Building on these insights, we propose *Detail Guidance* (DG), a MAs-driven, training-free self-guidance strategy to explicitly enhance local detail synthesis. Our DG can be seamlessly combined with CFG, enabling joint enhancement of detail fidelity and prompt alignment. Extensive experiments demonstrate the effectiveness of our approach in improving fine-grained detail synthesis.

ACKNOWLEDGEMENTS

The paper is supported in part by the National Natural Science Foundation of China (No. 62325109, 62561160155, 62595733), and in part by the Shanghai 'The Belt and Road' Young Scholar Exchange Grant (24510742000). Mehrtash Harandi is supported by the Australian Research Council (ARC) Discovery Program DP250100262.

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

## A DIFFUSION TRANSFORMER ARCHITECTURE

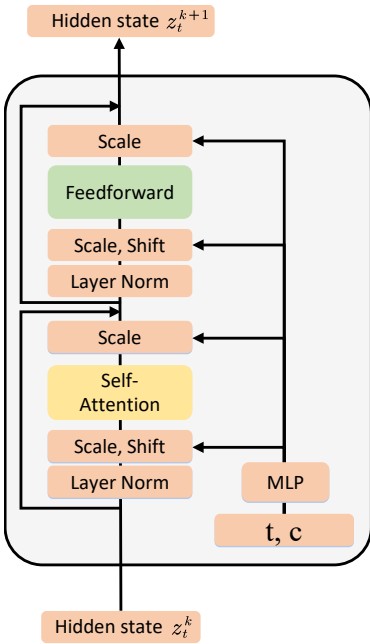

Figure 10: **Illustration of the architecture of DiT block** $D_k$**.**

We present the architecture of a DiT block in Figure 10. Each block consists of three key components: an AdaLN layer, a Self-Attention layer, and a Feedforward layer. The AdaLN layer encodes the input timestep $t$ and the additional conditioning information $c$ (e.g., class or text embedding) into channel-wise scale and shift parameters $\gamma_k$ and $\beta_k$. It then performs Adaptive Layer Normalization (AdaLN) on the hidden state $z_t^k$:

$$\hat{z}_t^k = \left(1 + \gamma_k\right) \operatorname{LayerNorm}(z_t^k) + \beta_k, \tag{6}$$

where $\gamma_k, \beta_k$ are regressed by the MLP networks of AdaLN layer conditioned on input timestep embedding $t$ and text embedding $c$:

$$\gamma_k, \beta_k, \alpha_k = \operatorname{MLP}_k(t, c) \tag{7}$$

where $\alpha_k$ scales the $k$-th residual connection.

Next, $\hat{z}_t^k$ is processed by the Self-Attention layer to produce an intermediate feature representation. A second adaptive layer normalization is then applied before passing this intermediate feature to the Feedforward layer, which outputs the updated hidden state. Finally, a residual connection combines the input and the transformed features to produce the block output:

$$z_t^{k+1} = z_t^k + \alpha_k\, D_k(z_t^k, t, c), \tag{8}$$

## B    LAYER PROPERTIES OF MASSIVE ACTIVATIONS

In this section, we examine the layer-wise characteristics of MAs in SD3, SD3.5, and Flux. The results are shown in Figures 11 to 13, revealing that massive activations consistently occur throughout all layers in these DiT models. Furthermore, it can be observed that the MAs dimensions remain consistent across all DiT layers (e.g., 810 for SD3, 676 for SD3.5), as shown in Figure 14.

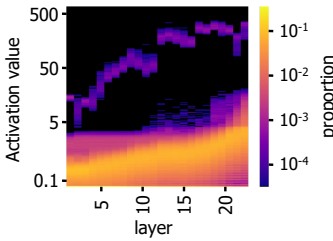

Figure 11: **Layer properties of MAs in SD3.** Massive activations in SD3 occur in all layers.

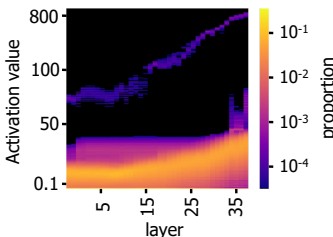

Figure 12: **Layer properties of MAs in SD3.5.** Massive activations in SD3.5 occur in all layers.

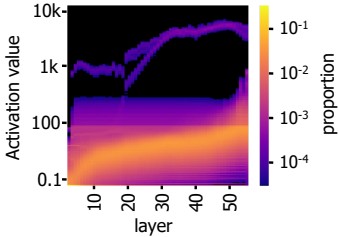

Figure 13: **Layer properties of MAs in Flux.** Massive activations in Flux occur in all layers.

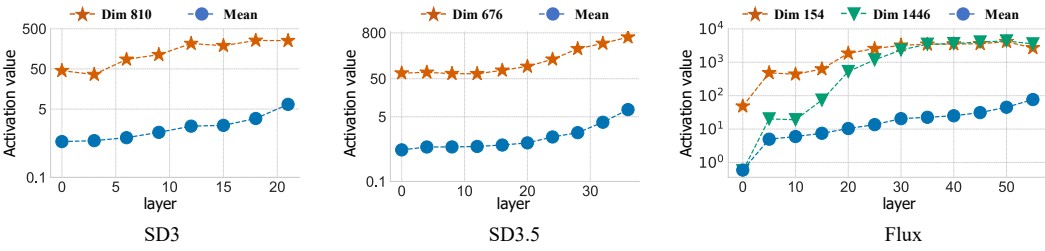

Figure 14: **MA dimensions are consistent across all layers.** We report the average activation values of MA dimensions across DiT layers. "Mean" denotes the mean activation value of the hidden state $z_t^k$. The results show that MA dimensions remain consistent across all layers for different DiT models (e.g., 810 for SD3, 676 for SD3.5).

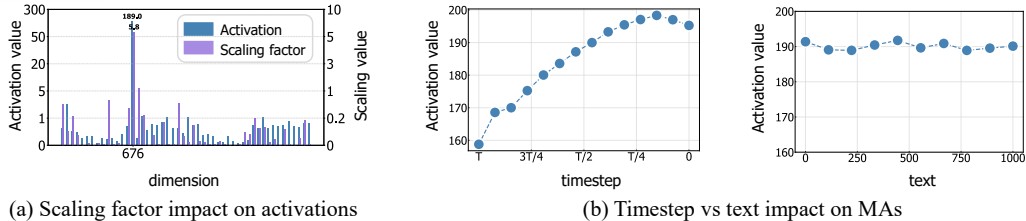

Figure 15: **Impact of the input timestep and text on Massive Activations (MAs) in SD3.5.** The input timestep $t$ plays a dominant role for massive activation: the magnitude of massive activations increases steadily as $t$ decreases from $T$ to 0.

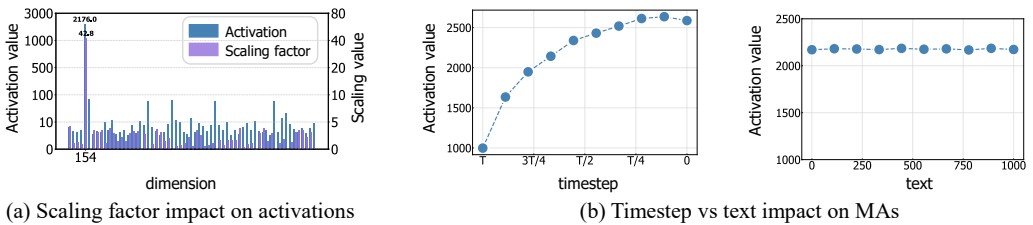

Figure 16: **Impact of the input timestep and text on Massive Activations (MAs) in Flux.** The input timestep $t$ plays a dominant role for massive activation: the magnitude of massive activations increases steadily as $t$ decreases from $T$ to 0.

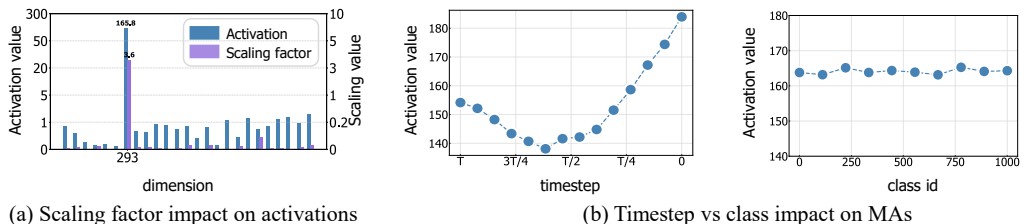

Figure 17: **Impact of the input timestep and text on Massive Activations (MAs) in DiT-XL.** The input timestep $t$ plays a dominant role for massive activation: the magnitude of massive activations increases steadily as $t$ decreases from $T/2$ to 0.

## C  TIMESTEP VS TEXT IMPACT ON MASSIVE ACTIVATIONS

We provide additional analysis for the massive activation of SD3-5, Flux and DiT-XL in Figures 15 to 17. As the hidden states $z_t^{k+1}$ in DiTs are computed via a residual connection (Equation (2)):

$$z_t^{k+1} = z_t^k + \alpha_k D_k(z_t^k, t, c) \tag{9}$$

where the $\alpha_k$ is the residual scaling factor. We first examine the impact of the residual scaling factor on these activations. Specifically, we visualize the average magnitude of each dimension for activation values and scaling factor values. We observe that the residual scaling factors $\alpha_k$ govern the dimension and values of massive activations in DiTs.

Furthermore, the residual scaling factor $\alpha_k$ is regressed by the AdaLN layer conditioned on input timestep embedding $t$ and text embedding $c$:

$$\alpha_k = \text{MLP}_k(t, c) \tag{10}$$

Therefore, we next investigate the respective impact of the input timestep $t$ and text $c$ to the massive activations. As shown in Figure 15(b), Figure 16(b) and Figure 17(b), it can be found that the magnitude of MAs is mainly influenced by the input timestep embedding while the input text embedding exerts minimal impact on it.

# D  ACTIVATION INTERVENTION FOR DITS

To better understand the functional role of Massive Activations (MAs) in the visual generation of Diffusion Transformers (DiTs), we conduct an activation intervention study with four experimental settings:

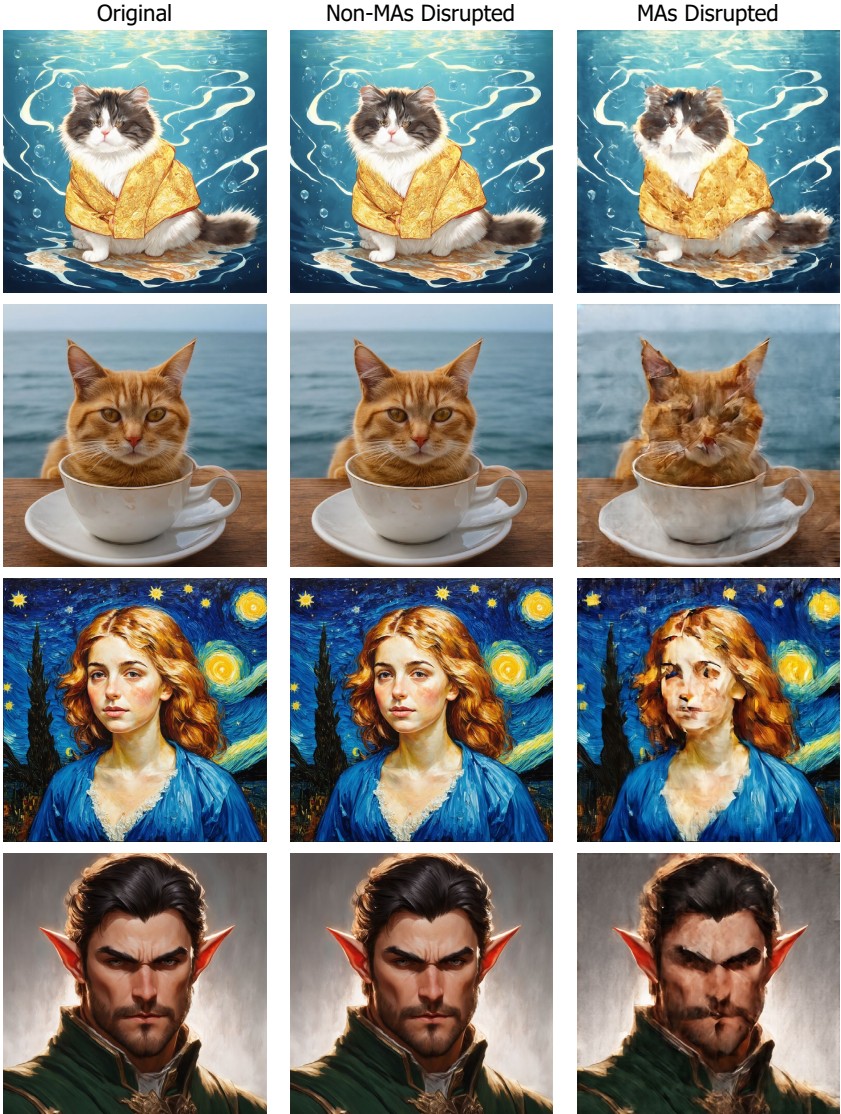

Figure 18: **Visual comparison of activation intervention.** MAs-disrupted models produce images with noticeably degraded local details, whereas non-MAs disruption preserves similar high-quality details with the original outputs. These results demonstrate that massive activations are crucial for fine-grained local detail synthesis in DiT generation process.

- **Original.** The pretrained DiT models are used to generate visual outputs without any modification.

- **MAs Disrupted.** We disrupt the massive activations by masking (zeroing out) their corresponding dimensions (e.g., dimension 293 for SD3 in Figure 2), as MAs consistently occur at fixed dimensions across all spatial tokens. Specifically, we mask the massive-activation dimensions in the block-output hidden state of a single block and propagate the modified state through the remaining DiT blocks. All other configurations (e.g., sampling steps) are kept same to the Original setting to ensure fair comparison.

- **Non-MAs Disrupted.** To provide a rigorous control, we additionally mask an equal number of randomly selected non-Massive dimensions instead of the massive-activation dimensions. This setting verifies that any observed effect arises specifically from disrupting MAs rather than from the masking operation itself.

- **Matched-non-MAs Disrupted.** To incorporate a stricter control, we introduce a Matched non-MAs disruption setting where the total perturbation applied to non-MA dimensions is adjusted to match that of the MAs-disruption setting. Specifically, we scale the masking of randomly selected non-MA dimensions so that the resulting L1 perturbation norm equals that of the MA disruption. In the SD3 model, achieving this match requires applying the scaled masking to approximately 50× non-MA dimensions to reach the same perturbation magnitude of MA.

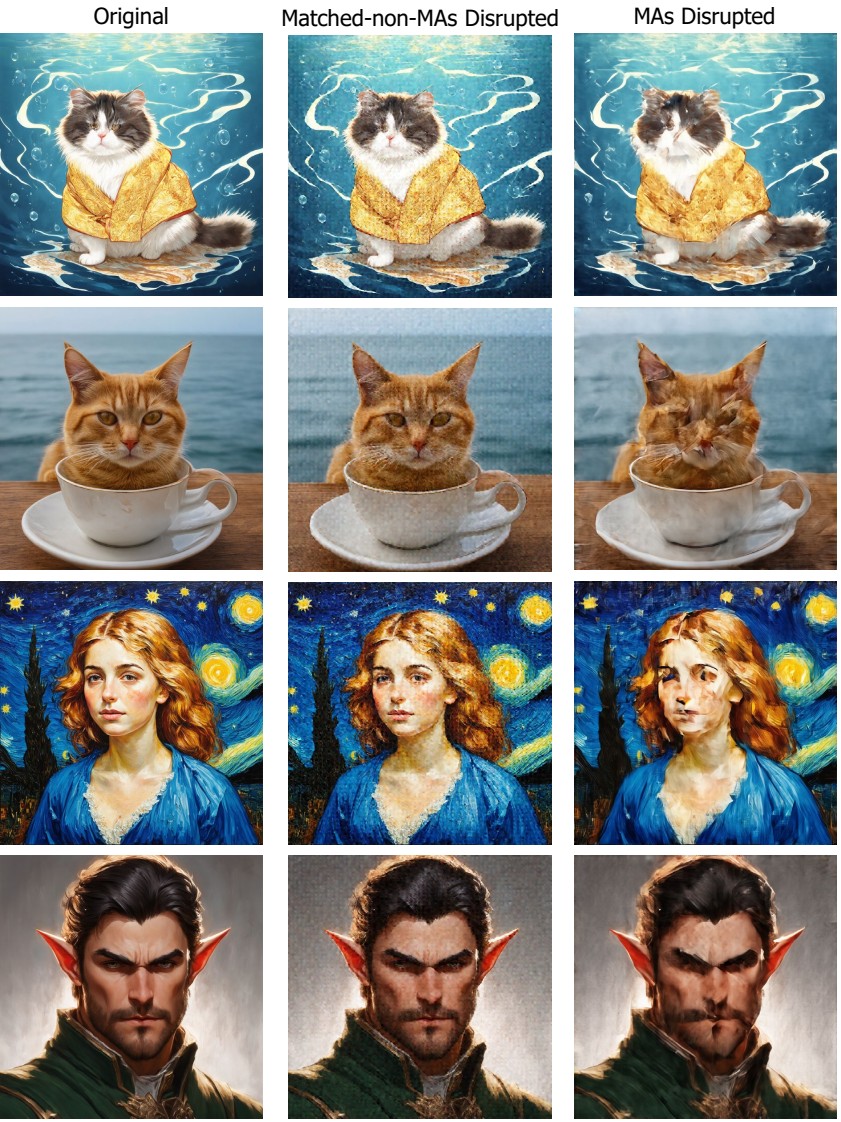

Figure 19: **Visualization of Matched-non-MAs disruption.** The Matched-non-MAs disruption preserves high-quality details similar to the original outputs, but introduces uniformly distributed noise artifacts. This happens because zeroing a large set of non-MA dimensions causes a systematic shift in the hidden-state distribution, disrupting the numerical balance of the decoding process. In contrast, disrupting MAs does not introduce such noise, as MAs act as modulation signals that guide the synthesis of local details for each token. Disrupting MAs primarily suppresses the model's ability to synthesize local details without affecting the overall decoding stability.

From the results in Figure 18, it can be observed that disrupting the massive activations in DiTs markedly degrades the fidelity of fine-grained local details, whereas disrupting non-MA dimensions has almost no effect on the generated images. Even under the stricter Matched-non-MAs Disrupted setting (see Figure 19), the generated outputs still preserve the similar high-quality semantic structure and fine-grained detail.

In the Matched-non-MAs Disrupted setting, the generated images exhibit uniformly distributed noise artifacts. This arises because zeroing out a large set of non-MA dimensions causes a systematic shift in the hidden-state distribution, which disrupts the numerical balance of the subsequent decoding process. As a result, isolated noise artifacts appear in the final images. In contrast, disrupting MAs causes a more focused degradation of local details without introducing such noise artifacts. This suggests that removing MAs does not alter the model's decoding stability but instead suppresses the modulation signal responsible for synthesizing local details. The absence of noise in the MA disruption setting further emphasizes the specific role of MAs in driving local detail synthesis. These findings demonstrate that massive activations play a crucial role in driving the synthesis of local details during the visual generation process of DiTs.

## E  SPATIAL ANALYSIS OF MASSIVE ACTIVATIONS

In this section, we provide a detailed spatial analysis of the Massive Activations (MA) in DiTs.

**Statistics of MA spatial map.** We first analyze the statistics of the spatial map in the MA dimension (dimension 810 for SD3). As shown in Table 3, at timestep $t = \frac{15}{28}T$, the median activation value of the hidden state $z_t^k$ is 0.60, which is more than 100× smaller than the activation values in the MA dimension, where the minimum activation reaches 71.0. Furthermore, the activation values in the MA spatial map range from 71.0 to 136.0, indicating that these activations are consistently large across all spatial tokens without isolated outliers. These observations demonstrate that MAs systematically appear at a fixed dimension for all spatial tokens and exhibit similar functional behavior throughout the DiT generation process.

| timestep $t$ | Spatial map of MAs: $z_t^k\,[:, 810]\,, k = 6$ | | | | Median($z_t^k$) |
| --- | --- | --- | --- | --- | --- |
| | Min | Max | Mean | Std | |
| $t = \frac{25}{28}T$ | 72.0 | 135.0 | 87.4 | 7.6 | 0.57 |
| $t = \frac{15}{28}T$ | 71.0 | 136.0 | 89.5 | 9.6 | 0.60 |
| $t = \frac{5}{28}T$ | 70.0 | 138.0 | 86.4 | 8.8 | 0.60 |

Table 3: **Statistics of MA spatial map for SD3.**

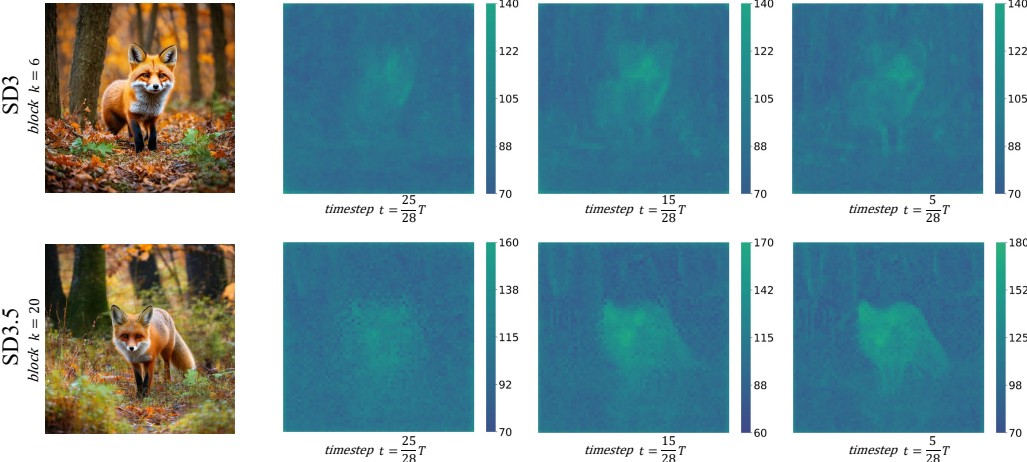

Figure 20: **Visualizations of MA spatial map**: MAs appear across all spatial tokens, while tokens corresponding to detail-richer regions (e.g., the fox) exhibit slightly higher MA values.

**Spatial map of MA dimension.** To analyze the spatial characteristics of MA, we visualize the spatial maps of MA dimension across different timesteps, as shown in Figure 20. From the results, we obtain

the following observations: **(1) Activations in the MA dimension remain consistently large across all spatial locations.** For SD3.5, the MA values range from approximately 70 to 180 without any extreme outliers. These activation values are more than 100x larger than the median activation value of the hidden states (approximately 0.6 from Table 3). This confirms that MAs appear across all spatial tokens, which serves as token-wise modulation signal for local detail synthesis of each token. **(2) Tokens corresponding to detail-richer regions exhibit slightly higher MA values.** In particular, the region containing the fox shows marginally higher MA values. This suggests that while the MA dimension is activated for every spatial token, its magnitude adapts subtly to local visual complexity. Spatial tokens responsible for synthesizing richer or more intricate details receive slightly stronger MA responses.

## F ADAPTING DG TO OTHER SCHEMES

Our DG strategy is a detail guidance approach driven by massive activations, and it is applicable to a wide range of models that exhibit massive activations, not limited to the standard DiTs. To assess the generality of our approach across different architectures, we evaluate our strategy on several other models, including efficient DiTs and non-DiT models.

**Efficient DiT.** To examine the robustness of DG on efficient DiT, we conduct experiments with the popular PixArt-alpha (Chen et al., 2023). PixArt-alpha adopts an efficient DiT architecture that injects text conditioning through cross-attention and employs **shared AdaLN layers** across all blocks to encode the timestep. These experiments allow us to specifically verify whether DG remains applicable under the shared-AdaLN setting.

**Non-DiT.** To further examine DG beyond transformer-based diffusion architectures, we experiment with the recent SANA (Xie et al., 2024), which introduces a novel Linear-DiT design. SANA replaces all vanilla attention with linear attention and integrates **3×3 convolutions** into the MLP. These experiments allow us to verify whether DG remains effective under such a hybrid design that incorporates convolutional operations

**Implementation details.** We analyze the pattern of Massive Activations (MAs) in the models and observe prominent MAs phenomena in both PixArt-alpha and SANA. These models exhibit similar characteristics to standard DiTs, with MAs primarily occurring across all spatial tokens of fixed dimensions. For PixArt-alpha, we create a "bad" version by masking dimension 273 in block 6. In the case of SANA, we mask dimensions 56 and 597 in block 2 to disrupt the MAs.

| Model | Type | DG | Prompt Alignment | | Detail Quality | |
|---|---|---|---|---|---|---|
| | | | Clipscore | Blipscore | HPSv2.1 | Aesthetic |
| PixArt-alpha | Cond | × | 22.64 | 68.41 | 25.63 | 6.01 |
| | | ✓ | **23.43** | **72.07** | **29.18** | **6.53** |
| | CFG | × | **26.20** | **87.64** | 29.99 | 6.21 |
| | | ✓ | 26.17 | 86.88 | **30.74** | **6.34** |
| SANA | Cond | × | 23.52 | 78.25 | 24.40 | 5.91 |
| | | ✓ | **24.98** | **84.11** | **28.72** | **6.12** |
| | CFG | × | 27.07 | **91.03** | 30.13 | 6.00 |
| | | ✓ | **27.20** | 90.25 | **30.52** | **6.07** |

Table 4: **Evaluation of DG on other architectures.** Our DG strategy generalizes effectively to other architectures and significantly enhances the visual detail quality for both Pixart-Alpha and SANA.

As shown in Table 4, our DG strategy consistently improves the performance of both PixArt-alpha and SANA. In the conditional generation setting, DG yields substantial improvements in detail quality, HPSv2.1 increases from 24.40 to 28.72 and the Aesthetic score from 5.91 to 6.12 for the Linear-DiT SANA model. Moreover, when combined with CFG, DG further enhances visual quality, improving HPSv2.1 from 30.13 to 30.52 and the Aesthetic score from 6.00 to 6.07. These results indicate that DG significantly boosts visual fidelity, with particularly notable gains in fine-grained local details. Overall, these findings demonstrate that DG generalizes effectively beyond standard DiT architectures, delivering robust improvements across both efficient DiT variants and non-DiT models.

# G    COMPUTATIONAL OVERHEAD OF DG

In this section, we investigate the computational overhead of our DG strategy. Specifically, we generate 100 images at 1024×1024 resolution using a single L40S (48GB) GPU. The average computational cost is reported in Table 5.

| Model | Type | GPU memory (GB) | Generation latency (s) | Aesthetic |
|---|---|---|---|---|
| SD3 | Cond | 17 | 2.3 | 5.58 |
| | CFG | 20 | 4.3 | 5.80 |
| | DG (Ours) | 20 | 3.5 | 6.01 |
| SD3.5 | Cond | 28 | 7.2 | 5.94 |
| | CFG | 32 | 15.7 | 6.01 |
| | DG (Ours) | 32 | 10.6 | 6.16 |
| Flux | Cond | 35 | 16.2 | 5.50 |
| | CFG | 42 | 36.0 | 5.96 |
| | DG (Ours) | 42 | 24.8 | 6.13 |

Table 5: **DG is approximately 1.5x faster than CFG.** Since DG only forwards one conditional branch before the disrupted depth $m$, while CFG requires both the conditional and unconditional branches to be processed, DG achieves superior efficiency. For the SD3.5 model, our DG generates a 1024x1024 image in 10.6s, approximately 1.5x faster than CFG, which takes 15.7s.

**DG is approximately 1.5× faster than CFG and delivers superior performance.** As demonstrated in Table 5, DG achieves higher performance than CFG (e.g., 6.16 vs. 6.01 on SD3.5 model). Moreover, DG generates a 1024×1024 image in 10.6s, approximately 1.5× faster than CFG, which requires 15.7s. This efficiency stems from DG's architectural design: it leverages an MAs-disrupted conditional branch to guide the base model. Before the disruption depth (e.g., $m = 20$ out of $N = 38$ blocks for SD3.5), DG requires forwarding only the conditional branch, while CFG necessitates forwarding both the conditional and unconditional branches throughout the entire sampling process. This architectural difference makes DG both more efficient and effective than CFG.

# H    MORE IMPLEMENTATION DETAILS

## H.1    IMPLEMENTATION DETAILS FOR FIGURES

This section provides the implementation details for the figures 1-5 presented in our paper.

**Figure 1.** Prompts used for the visual examples in the figure:

- **Prompts for the examples with Detail Guidance (DG).** [(1)`"Card Magic the gathering style of tom whalen022 2e SM Ricardo. Lavage du char au gazole a Biesheim."`, (2)`"Movie Still of The Joker wielding a red Lightsaber, Darth Joker a sinister evil clown prince of crime, HD Photograph"`, (3)`"Bichon maltais fou"`, (4)`"Frank Lloyd public library with a coffee shop, mid century, interior"`].

- **Prompts for the examples with DG and CFG.** [(1)`"A cat, chubby, very fine wispy and extremely long swirly wavy fur, under water, Kuniyoshi Utagawa, Hishida Shunsō, a very curvy chubby cat, golden embroidery fabric kimono, flowing glowing biomorphic wisps, phosphorescent swirls, tendrils, wavelets, streamers, a murmuration of bioluminescent bubbles, , detailed and intricate, elegant aesthetic, ornate, finely detailed, 128K UHD Unreal Engine, octane, fractal pi, fBm"`, (2)`"Dream alpine treehouse with sweeping mountain views"`]

**Figures 2 to 5.** For configuration of the hidden states (block index $k$ and timestep index $t$) used for different DiT models, please refer to Table 6.

| | Figure 2 | | | | Figure 3 | Figure 4 | Figure 5 |
|---|---|---|---|---|---|---|---|
| | DiT-XL/2 | SD3 | SD3.5 | Flux | | | |
| **Architecture** | | | | | | | |
| Image size | 256x256 | | 1024x1024 | | 256x256 | 1024x1024 | 1024x1024 |
| Hidden size $d$ | 1152 | 1536 | 2432 | 3072 | - | 1536 | 1536 |
| **Hidden state $z_t^k$** | | | | | | | |
| Total blocks $N$ | 28 | 24 | 38 | 57 | - | 24 | 24 |
| Block index $k$ | 14 | 12 | 19 | 28 | - | 12 | 6 |
| Total timesteps $T$ | 250 | 28 | 28 | 50 | 250 | 28 | 28 |
| Timestep index $t$ | 125 | 14 | 14 | 25 | 125 | 14 | - |

Table 6: **Configuration of hidden states used in Figures 2-5.**

## H.2 IMPLEMENTATION DETAILS OF EXPERIMENTAL RESULTS

We implement DG on three pretrained large diffusion models: **SD3-Medium** (Esser et al., 2024), **SD3.5-Large** (Esser et al., 2024), and **Flux-dev** (black-forest labs, 2024). Notably, **Flux** is a CFG-distilled model. To evaluate DG independently of CFG, we adopt the *de-distilled* variant from (black-forest labs, 2024). Full experimental settings are provided in Tables 7 and 8.

**Configurations for DG.** For each Diffusion Transformer, we construct a degraded *detail-deficient* model $D_{\theta,m}$ by disrupting the dimensions corresponding to massive activations in the $m$-th blocks, following the intrinsic activation patterns of each DiT. Detailed configurations are summarized in Table 7.

**Hyperparameters setup.** All models adopt the default diffusion sampling settings (e.g., sampler type and number of steps). Specific hyperparameter choices are listed in Table 8.

**Computing Resources.** All experiments are performed on a single NVIDIA L40S (48 GB) GPU. DG builds the degraded *detail-deficient* model by directly disrupting massive activations in hidden states **without additional training**.

| Model | Blocks N | Hidden size d | Disrupted dimensions | Disrupted depth $m$ |
|---|---|---|---|---|
| SD3 | 24 | 1536 | 810 | 6 |
| SD3.5 | 38 | 2432 | 676 | 20 |
| Flux | 57 | 3072 | [154, 1446] | 22 |

Table 7: **Configurations of Detail Guidance (DG) for different DiTs.**

| Model | Guidance Type | sampling step | $\lambda$ | $w$ |
|---|---|---|---|---|
| | CFG | 28 | 4 | - |
| SD3 | DG | 28 | - | 1 |
| | CFG+DG | 28 | 3 | 1 |
| | CFG | 28 | 4 | - |
| SD3-5 | DG | 28 | - | 4 |
| | CFG+DG | 28 | 3 | 2 |
| | CFG | 50 | 3.5 | - |
| Flux | DG | 50 | - | 4 |
| | CFG+DG | 50 | 3 | 2 |

Table 8: **Hyperparameter setup.**

### H.3 Evaluation Details

We evaluate different guidance strategies from two key perspectives: *prompt alignment* and *detail quality*. Prompt alignment reflects how well the generated image semantically matches the input prompt, while detail quality measures the fidelity and richness of fine-grained visual details.

Specifically, we adopt Clipscore (Radford et al., 2021) and Blipscore (Li et al., 2022) to quantify prompt alignment, and employ HPSv2.1 (Wu et al., 2023) and Laion-Aesthetics (Schuhmann, 2022) as indicators of visual detail quality. The details of each metric are as follows.

**Clipscore** measures the global semantic consistency between text and image by computing the cosine similarity between their CLIP-encoded features. We adopt the *clip-vit-large-patch14* version for all experiments.

**Blipscore** estimates prompt-image alignment through a fine-grained image-text matching model (BLIP), capturing nuanced semantic relationships beyond global similarity. We use the *blip-itm-large-coco* version to evaluate the model.

**HPSv2.1** is a human preference score that provides a perceptual measure of visual realism and aesthetic quality. It is widely used to benchmark high-fidelity image synthesis, and we adopt HPSv2.1 for evaluation.

**Laion-Aesthetics** predicts aesthetic appeal using a model trained on LAION's large-scale human-rated dataset, serving as an automated proxy for human aesthetic assessment.

## I Class-conditional generation

To evaluate the robustness of our Detail Guidance (DG) strategy, we perform class-conditional generation on the ImageNet 256×256 dataset by applying DG to the pretrained DiT-XL/2 model (Peebles & Xie, 2023).

| Type | DG | FID ↓ | IS ↑ | Prec. ↑ | Rec. ↑ |
|------|-----|-------|-------|---------|--------|
| Uncond | × | 16.95 | 105.64 | 0.61 | 0.76 |
|        | ✓ | 9.68 | 122.22 | 0.66 | 0.67 |
| Cond | × | 9.52 | 122.79 | 0.66 | 0.63 |
|      | ✓ | 5.77 | 179.26 | 0.78 | 0.55 |

Table 9: **Performance comparison on dataset ImageNet** $256 \times 256$. Prec: Precision, Rec: Recall.

For DiT-XL/2, we set the disrupted depth $m = 7$. We assess DG under both unconditional and conditional generation settings, with results reported in Table 9. DG delivers consistent performance improvements in both settings, demonstrating the robustness of our guidance strategy.

## J User study

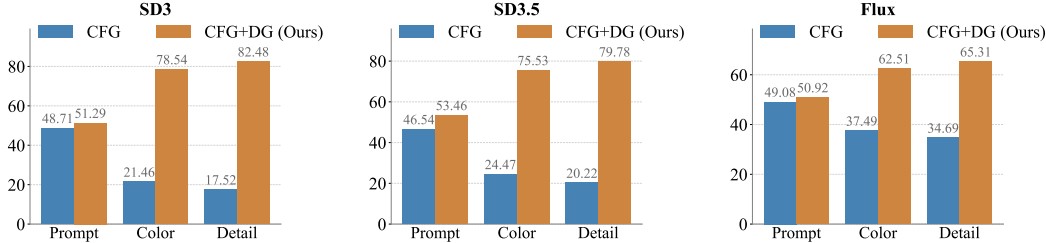

Figure 21: **User study on SD3, SD3.5, and Flux.** We report the win rates comparing CFG with our method.

We conduct a user study to evaluate the benefits of our DG strategy from three key aspects: **prompt alignment**, **detail preservation**, and **color consistency**. Prompt alignment measures how well the

generated images match the input prompts. Detail preservation reflects the fidelity of fine-grained visual details, while color consistency captures the naturalness and realism of colors.

For each model, 20 annotators compared 100 pairs of images produced by **CFG** and **CFG + DG** with respect to these criteria. We report the averaged win rates in Figure 21, which show that our approach yields significant improvements across all metrics, particularly for *Detail* and *Color*.

## K    DISCUSSION: CFG AND AUTOGUIDANCE VS. OUR DG

To facilitate a clearer conceptual understanding of DG and its relationship to existing guidance strategies, we present a unified formulation of CFG, Autoguidance, and DG in Table 10.

| Type | Formulation | Disruption |
|---|---|---|
| CFG | $\hat{D}_\theta\left(z_t, t, c\right) = D_\theta\left(z_t, t, c\right) + w\left(D_\theta\left(z_t, t, c\right) - D_\theta\left(z_t, t, \hat{c}\right)\right)$ | $c$ |
| Autoguidance | $\hat{D}_\theta\left(z_t, t, c\right) = D_\theta\left(z_t, t, c\right) + w\left(D_\theta\left(z_t, t, c\right) - D_{\theta^*}\left(z_t, t, c\right)\right)$ | $\theta$ |
| DG | $\hat{D}_\theta\left(z_t, t, c\right) = D_\theta\left(z_t, t, c\right) + w\left(D_\theta\left(z_t, t, c\right) - D_\theta\left(\hat{z}_t, t, c\right)\right)$ | $z_t$ |

Table 10: **Unified formulations of CFG, Autoguidance and Our DG.** $\hat{c}$ denotes the unconditional prompt, $\theta^*$ denotes under-capability model, and $\hat{z}_t$ denotes $z_t$ with MAs-disrupted hidden state.

Under this unified formulation (see Table 10), the distinction becomes clear:

- **CFG** constructs a degraded version by "disrupting" the input prompt condition $c$, resulting in prompt-alignment guidance.

- **Autoguidance** constructs a degraded version by "disrupting" the model $\theta$, resulting in entangled prompt-alignment and visual detail guidance.

- **Our DG** constructs a degraded version by "disrupting" the visual input $z_t$ itself through MA disruption, resulting in visual detail guidance.

Our DG can be seamlessly combined with CFG, enabling a decoupled guidance mechanism where CFG handles prompt alignment while DG focuses on visual detail guidance.

## L    ANALYSIS OF FAILURE CASES

In this section, we present visualizations and analyses of the failure cases associated with our DG strategy. DG is explicitly designed to enhance the local detail fidelity of generated images. However, this emphasis on fine-grained detail can occasionally compromise semantic faithfulness. In particular, when the prompt specifies strong stylistic, identity-related, or conceptual requirements, DG may favor detail enhancement over strict adherence to the intended semantics.

As illustrated in Figure 22, DG produces outputs with noticeably richer textures and enhanced local details but may fail to fully satisfy the semantic requirements of the prompt. For instance, given the prompt *"The Mona Lisa as a vogue model, 1989 punk-inspired portrait, dramatic lighting, cinematic lighting"*, DG generates a portrait with improved local details and textures, yet the output lacks the expected *vogue* style and the distinctive identity features of *Mona Lisa*. A promising direction to address this limitation is to combine DG with CFG, enabling joint control over both local detail and semantic alignment. Moreover, when the guidance scale becomes relatively large (e.g., greater than 5), our DG strategy may also exhibit oversaturation effects, similar to those observed with CFG.

## M    LIMITATIONS AND FUTURE WORK

**Limitations**. Our Detail Guidance (DG) method is primarily designed to explicitly enhance the detail quality of generated images. While it does improve prompt alignment, its ability to ensure strong alignment throughout the generation process is somewhat constrained. For prompts that specify strong stylistic, identity-related, or conceptual requirements, we recommend combining DG with CFG to provide joint guidance on prompt alignment and detail enhancement.

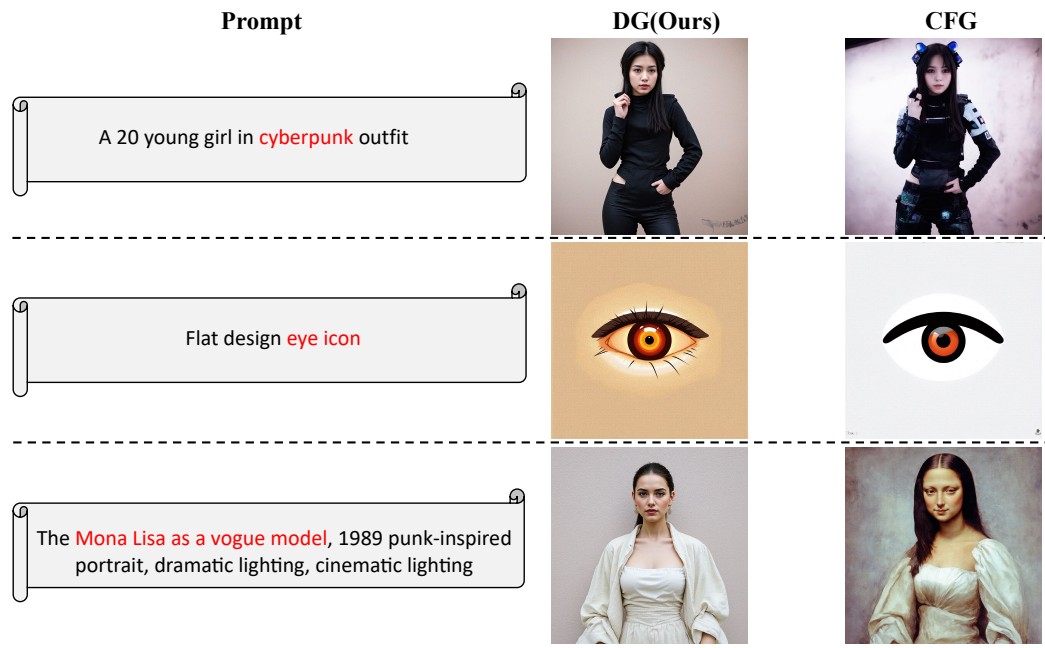

Figure 22: **Failure Cases of Our DG.** DG explicitly enhances local detail fidelity, which can occasionally compromise semantic alignment. DG produces outputs with improved texture and fine-grained details but fails to satisfy the prompt requirement (e.g., "Mona Lisa as a Vogue model"), leading to missing identity and stylistic attributes. The words highlighted in red correspond to semantic requirements in the prompt that are not fulfilled by the generated image from DG.

**Future Work.** This paper mainly explores the potential of Massive Activations (MAs) during the sampling stage of DiTs. A more promising direction for future work lies in leveraging the capacity of MAs during the training stage. Since MAs play a crucial role in local detail synthesis, incorporating them more effectively into the training process could significantly enhance the performance of DiTs. This approach offers valuable insights for optimizing future DiTs, driving further advancements in their generative capabilities.

# N  USE OF LARGE LANGUAGE MODELS (LLMS)

In preparing this manuscript, we used large language models solely as a lightweight writing aid for grammar, wording, and formatting suggestions. The models were *not* used to generate research ideas, design algorithms, write code, run experiments, analyze data, or draft scientific content. All technical claims, methods, and conclusions were conceived, produced, and verified by the authors. Suggested edits from LLMs were manually reviewed and integrated at the authors' discretion. And we accept full responsibility for the accuracy and integrity of the manuscript, including ensuring that no plagiarized or misrepresented content from a LLM is included.

## O    MORE QUALITATIVE RESULTS FOR INTEGRATION WITH CFG

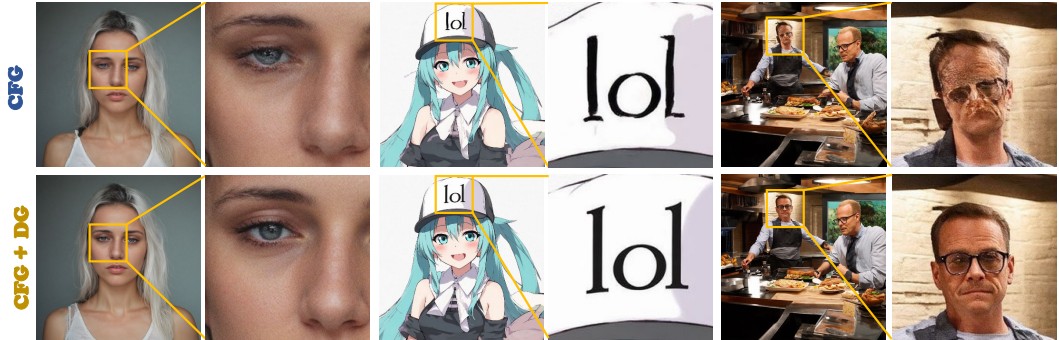

Figure 23: **Visual results on SD3.**

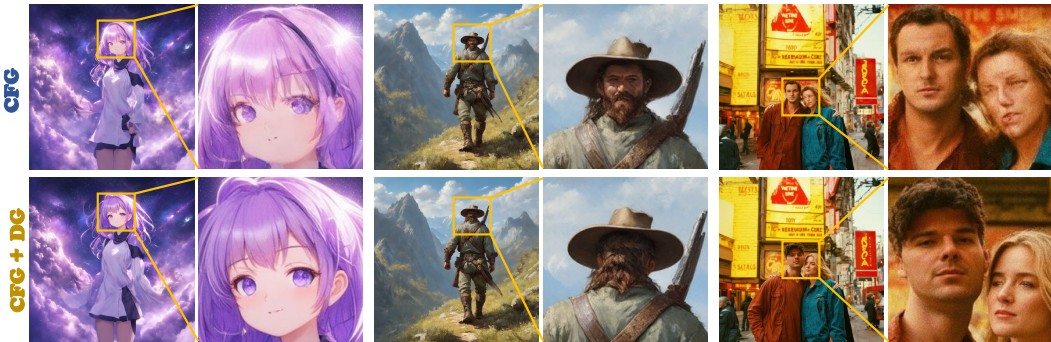

Figure 24: **Visual results on SD3.**

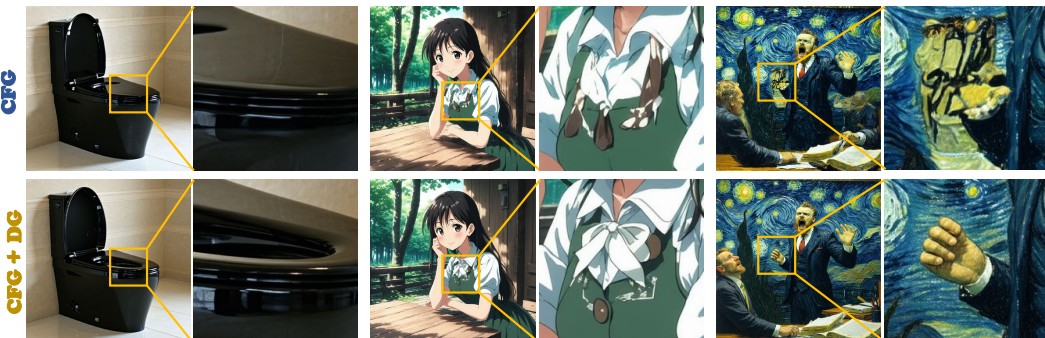

Figure 25: **Visual results on SD3.5.**

## P    MORE QUALITATIVE RESULTS FOR DG

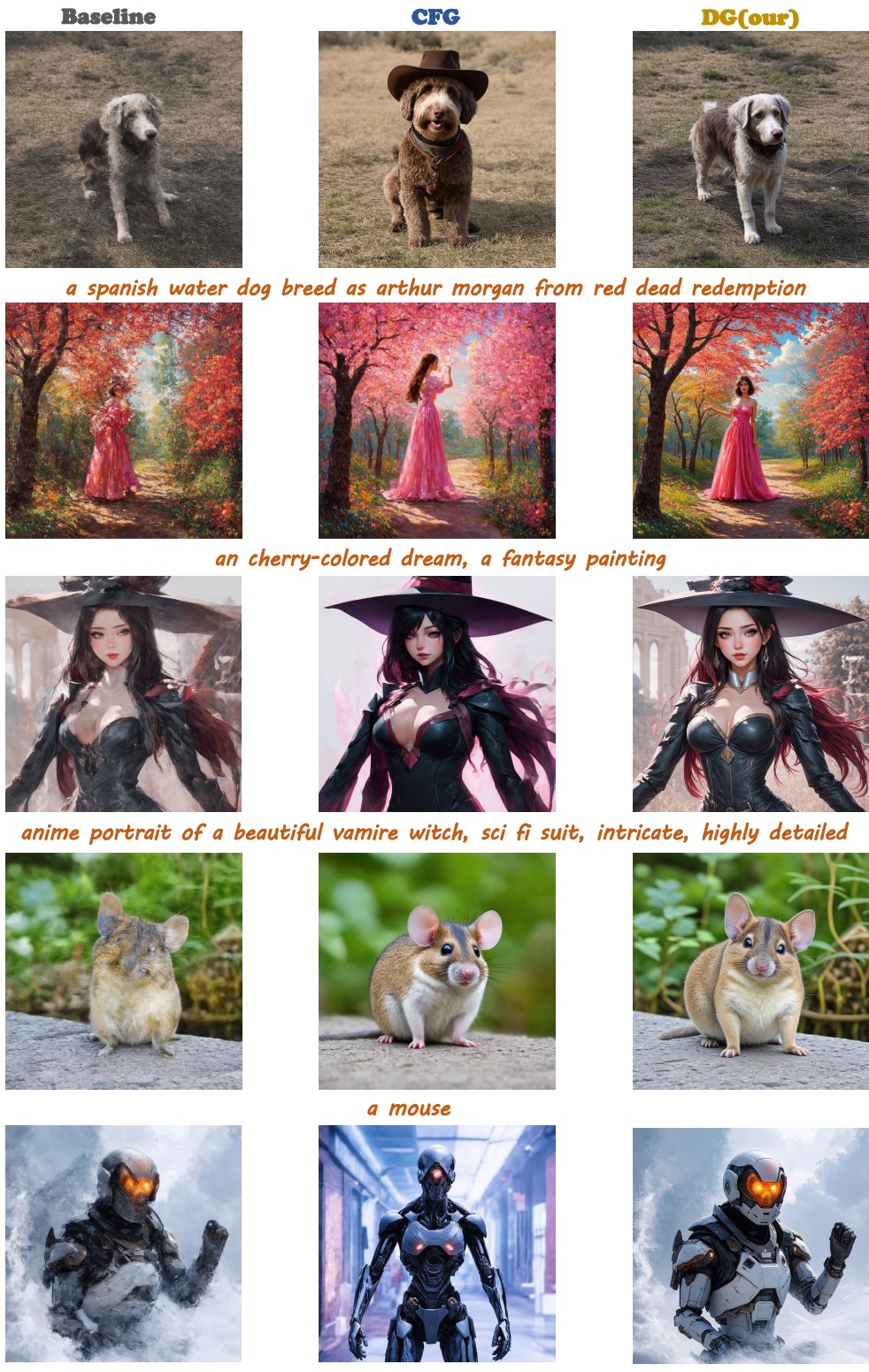

Figure 26: **Visual results on SD3.** Baseline indicates visual output without CFG.

Figure 27: **Visual results on SD3.5.** Baseline indicates visual output without CFG.

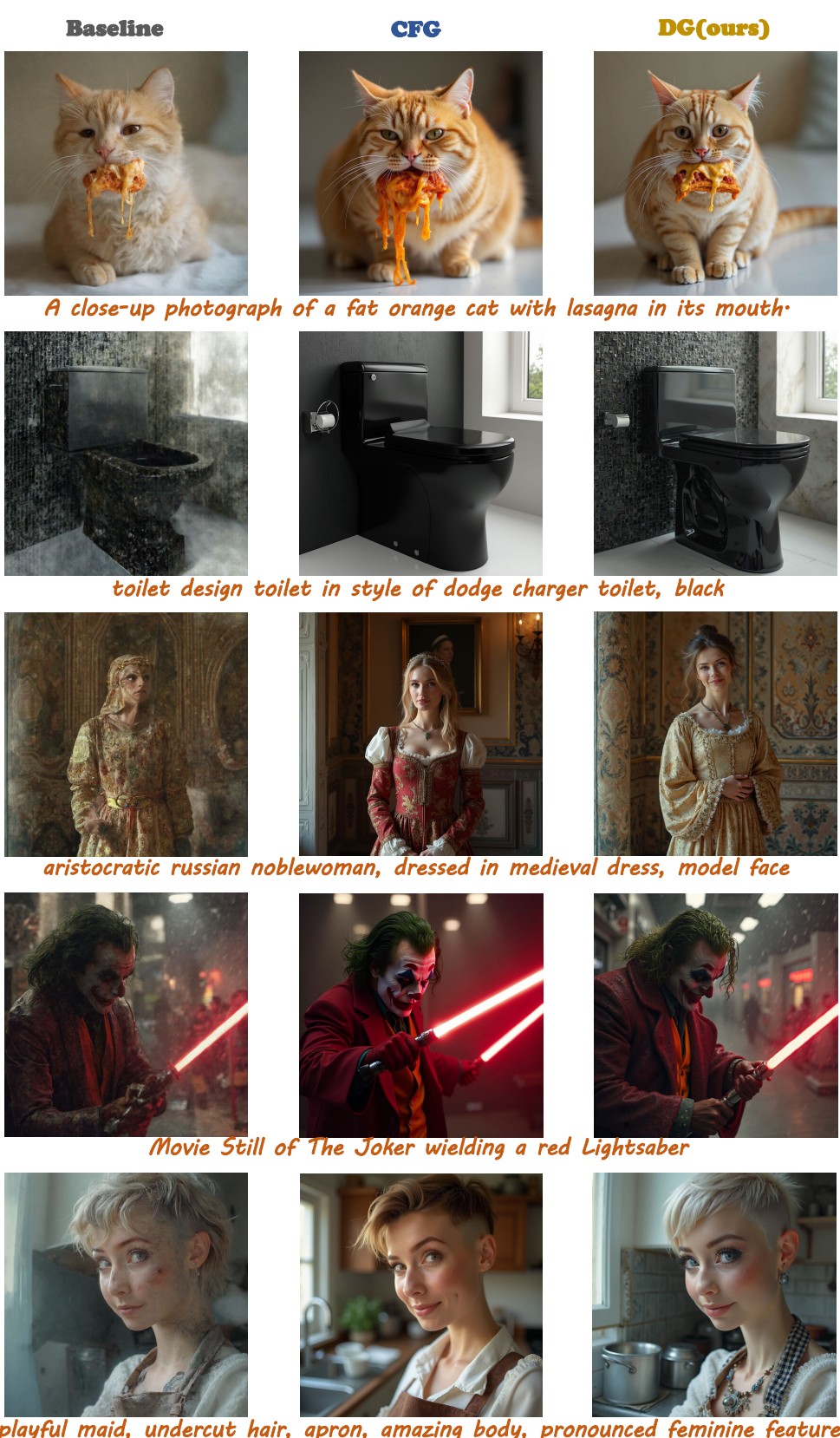

Figure 28: **Visual results on Flux.** Baseline indicates visual output without CFG.

