# OpenReview forum: "Massive Activations are the Key to Local Detail Synthesis in Diffusion Transformers"
_ICLR.cc/2026/Conference — ICLR 2026 Poster_

### Official Review · Reviewer_SiyV · 2025-10-29

**Soundness:** 3
**Presentation:** 4
**Contribution:** 3
**Rating:** 4
**Confidence:** 4

**Summary:**

The authors study the role of massive activation (MA) in DiT models. Massive activation is a common observation in transformer models. The authors attribute massive activations in DiT to the generation of local detail. Experiments show that disruption of MA degrades local information. The authors propose Detail Guidance (DG) to enhance CFG on local details. The manuscript is clearly written. The analysis of MA is comprehension. The experimental design is legit. Overall, it is a high-quality study. Though, I note several unclear parts below and suggest several controls to support the effectiveness of DG.

**Strengths:**

Massive activation is a common observation in transformer models across modalities. Its functional role is less known in DiT models. The analysis provided by the study is comprehensive (Figure 3-5). The finding of MA’s role of synthesizing is novel and supported. Leveraging this finding, Detail Guidance seems to be an effective, though simple, way to enhance local details.

**Weaknesses:**

**DG control**

As the authors mentioned, Karras et al 2024 proposed using an under-trained version to contrast the condition path. DG works in a similar way. Method in Karras et al 2024 should be used as a control. Or, if MA-disruption is really only about disrupting local detail features, there should be another control that uses blurred conditional path as the unconditional signal.

**MA-disruption control**

In “Non-MA disruption”, it is hard to say that zeroing non-MA dims is a fair control. The amount of perturbation in “Non-MA disruption” is smaller than in “MA disruption”. To be a fair control, the same total magnitude of disruption should be the same.

**Missing information, or not emphasized in the main text**

Are MA studied in the conditional path (D_/theta(z,t,c) in eq 1) or unconditional path (D_/theta(z,t) in eq 1)? From eq. 2, it seems the state is taken from the unconditional path, but in eq 4, the MA is studied/manipulated in the conditional path. The interpretation of the results heavily depends on where MA are found and manipulated. Authors should clarify where the hidden states are taken from in Figure 2-5.

Figure 2 misses information on which layer and time step the activations are taken from.

**Questions:**

**Spatial map of MA**

In the ViT studies, the MA often appears in only several background tokens in the image. If the role of MA in DiT is to synthesize details, then does it mean the MA is supposed to be in all the tokens, or only in tokens with plenty of details? If only disrupting several tokens’ MA, is the details only missing locally? The special activation map of MA dim is not shown. Showing an activation map of MA dim would be helpful to understand its actual role.

---

> ### Author Response · Authors · 2025-11-21
> **Response to Reviewer SiyV (1/3)**
>
> Thanks for your valuable suggestions. We’re glad that you found our work insightful and interesting. Please see the responses to your comments. In the revised draft, we mark our major revisions as “blue”.
>
> ### **Q1. DG control**
> Thank you for your valuable comments. We would like to kindly clarify that our DG use the same control mechanism as Autoguidance (Karras et al., 2024) [1], where we use the **blurred (MAs-disrupted) conditional path serves as the unconditional signal**, as shown in Eq. 4. We sincerely apologize for the inaccuracies in Eqs. 2 and 3 in the previous draft, and we have corrected them and related content in the revision (revising $D_k(z_t^k, t)$ to $D_k(z_t^k, t, c)$).
> Conceptually, our design follows the control mechanism of Autoguidance (Karras et al., 2024): Autoguidance constructs an unconditional signal by using an under-trained conditional branch, while our DG constructs the unconditional signal by **applying MA disruption to the conditional branch**. By discrupting massive activations, we obtain a degraded conditional path that produces detail-deficient outputs. This degraded path then guides the original conditional path toward higher-quality detail synthesis.
>
>
> ### **Q2. MA-disruption control**
>
> Thank you for your insightful comment. We fully agree with you that “the amount of perturbation in Non-MA disruption is smaller than in MA disruption.” We would like to clarify our motivation and experimental design from two perspectives:
>
> **(1) MA disruption naturally introduces larger perturbation.**
>
> Massive Activations inherently exhibit much larger magnitudes than Non-massive activations. This is precisely why we focus on them: their unusually large values exert a disproportionately strong influence on the hidden state and thus likely play a critical role in image synthesis. In our original control setup, we ensured that the number of perturbed MA and Non-MA dimensions was same so that both conditions were structurally comparable.
>
> **(2) Matched-non-MAs disruption.**
>
> Following your suggestion, we add a stricter control experiment in which the total perturbation strength applied to non-MA dimensions is matched to that of MA disruption (Appendix D).
>
> **Experimentl setting.** In this Matched-non-MAs disruption setting, we scale the masking of randomly selected non-MA dimensions such that the resulting **L1 perturbation norm equals that of the MA disruption**. For the SD3 model, achieving this requires perturbing approximately 50× non-MA dimensions to match the perturbation magnitude produced by zeroing the MA dimension.
>
> **Experimentl results.** From the results in Figure 19 of Appendix D, we can obtain the following observations.
> - **MAs disruption leads to siginificant detail degradtion while Matched-non-MAs not.** As shown in Figure 19, even under the stricter Matched-non-MAs Disrupted setting, the generated outputs still preserve the similar high-quality semantic structure and fine-grained detail. In contrast, the MAs-disrupted models produce images with noticeably degraded local details. These results demonstrate that massive activations are crucial for visual detail synthesis in DiT generation process.
> - **Matched-non-MAs disruption introduces uniform noise artifacts while MAs not.** In the Matched-non-MAs Disrupted setting (see Figure 19), the generated images exhibit uniformly distributed noise artifacts. This arises because zeroing out a large set of non-MA dimensions causes a systematic shift in the hidden-state distribution, which disrupts the numerical balance of the subsequent decoding process. As a result, isolated noise artifacts appear in the final images. In contrast, disrupting MAs causes a more focused degradation of local details without introducing such noise artifacts. This suggests that removing MAs does not alter the model’s decoding stability but instead suppresses the **modulation signal responsible for synthesizing local details**. The absence of noise in the MA disruption setting further emphasizes the specific role of MAs in driving local detail synthesis.
>
> These findings demonstrate that massive activations play a crucial role in driving local detail synthesis during the visual generation process of DiTs. In the revised draft, we have incorporated these additional experiments and analyses into the Appendix D.
>
> [1] Guiding a Diffusion Model with a Bad Version of Itself, NeurIPS 2024.

---

> ### Author Response · Authors · 2025-11-21
> **Response to Reviewer SiyV (2/3)**
>
> ### **Q3. Missing information, or not emphasized in the main text**
> Thank you for your valuable comments and suggestion.
>
> **(1) We study the MA in the conditional path ($D_\theta(z_t,t,c)$ in eq 1).** For our DG strategy, we use the blurred (MAs-discrupted) conditional path serves as the unconditional signal. We sincerely apologize for the inaccuracies in Eqs. 2 and 3 in the previous draft, and we have corrected them and related content in the revision (revising $D_k(z_t^k, t)$ to $D_k(z_t^k, t, c)$). .
>
> **(2) Details for Figure 1-5.** We appreciate the request for clarification. For Figure 2, we visualize the hidden state $z_t^k$ from the middle block ($k = N/2$) and timestep ($t=T/2$). We have revised the caption of Figure 2. The revised caption of Figure 2 is presented as follows:
> > **Massive Activations in DiTs.** The activation magnitudes of internal hidden states from the middle block ($k = N/2$) and timestep ($t=T/2$). We present the average magnitudes over 1,000 text prompts. Massive Activations (MAs) are consistently concentrated in a few fixed dimensions across all image patch tokens. The MA dimensions remain consistent across all layers (see Figure 14).
>
> Moreover, in the revised draft, we have provided comprehensive implementation details for Figures 1–5, including **the prompts and the hidden state configurations**, as shown in Appendix H.1 and Table 6.
> ### **Q4. Spatial map of MA**
> Thank you for your insightful comments.
> To better clarify the functional role of Massive Activations (MA), we first elaborate on our hypothesis that *"during training, DiT learns to assign massive activations to all spatial tokens to drive fine-grained local detail synthesis of each token"*. During training, every spatial token must be optimized to generate its own local details. To facilitate this process, DiT learns to allocate MA to each token, which acts as a **token-wise modulation signal that drives local detail synthesis of corresponding token**. As a result, MAs consistently appear at fixed dimensions across all spatial tokens, functioning as a modulation mechanism that drives the local detail generation of each token.
>
> Next, we further validate and elaborate on the role of MAs from the following perspectives.
>
> **(1) Statistics of MA spatial map.**
>
> We first analyze the statistics of the spatial map in the MA dimension (dimension 810 for SD3). As shown in *Table i*, at timestep $t = \frac{15}{28}T$, the median activation value of the hidden state $z_t^k$ is 0.60, which is more than 100× smaller than the activation values in the MA dimension, where the minimum activation reaches 71.0. Furthermore, the activation values in the MA spatial map range from 71.0 to 136.0, indicating that these activations are consistently large across all spatial tokens without isolated outliers. These observations demonstrate that MAs systematically appear at a fixed dimension for all spatial tokens and probably exhibit similar functional behavior throughout the DiT generation process.
>
>
> |timestep $t$|Min| Max |Mean|Std | Median($z_t^k$) |
> | ----- |:-----: | :-----: | :-------: |:-------: |:-------: |
> $t=\frac{25}{28}T$|72.0|135.0|87.4|7.6|0.57
> $t=\frac{15}{28}T$|71.0|136.0|89.5|9.6| 0.60
> $t=\frac{5}{28}T$|70.0|138.0|86.4|8.8|0.60
>
> *Table i. Statistics of MA spatial map for SD3. We use the hidden state $z_t^k$ from block k=6. The spatial map of MAs comes from MA dimension 810.*
>
> **(2) Spatial map of MA dimension.**
>
> To analyze the spatial characteristics of MA, we visualize the spatial maps of MA dimension across different timesteps, as shown in Figure 20 of Appendix E. From the results, we obtain the following observations:
> - **Activations in the MA dimension remain consistently large across all spatial locations.** For SD3.5, the MA values range from approximately 70 to 180 without any extreme outliers. These activation values are more than 100x larger than the median activation value of the hidden states (approximately 0.6 from *Table i*). This confirms that MAs appear across all spatial tokens, which serves as token-wise modulation signal for local detail synthesis of each token.
> - **Tokens corresponding to detail-richer regions exhibit slightly higher MA values.** In particular, the region containing the fox shows marginally higher MA values. This suggests that while the MA dimension is activated for every spatial token, its magnitude adapts subtly to local visual complexity. Spatial tokens responsible for synthesizing richer or more intricate details receive slightly stronger MA responses.
>
> Based on these findings, we extend our hypothesis regarding the role of MAs: *during training, DiT learns to assign Massive Activations (MAs) to all spatial tokens to drive local detail synthesis of each token, and allocate slightly higher MA magnitudes to the detail-richer tokens.* In the revised draft, we have added a new appendix section (Appendix E) to provide the spatial analysis for Massive Activations.

---

> ### Author Response · Authors · 2025-11-21
> **Response to Reviewer SiyV (3/3)**
>
> ### **Q5. If only disrupting several tokens’ MA, is the details only missing locally?**
>
> **We can achieve token-specific guidance using DG by selectively disrupting MAs in target tokens.**
>
> Our analysis has revealed that DiT assigns Massive Activations (MAs) to every spatial token, where these activations serve as token-wise modulation signals that drive the local detail synthesis of each token. To further investigate this locality property of MA, we design a **Local DG** experiment in which we mask MAs exclusively for a selected subset of spatial tokens (e.g., tokens corresponding to a specific object or region), rather than masking MAs across all spatial tokens as in the original DG setting (see Figure 9).
>
> As demonstrated in Figure 9, Local DG effectively guides local details in the generated images. Specifically, by disrupting the MAs only in the tokens corresponding to the dog, **Local DG successfully enhances the local details of the dog while leaving other regions, such as the cat and background, essentially unaffected**. These results clearly demonstrate the locality of massive activations: **each MA primarily drives the local detail synthesis of its corresponding spatial token**. By exploiting this property, we can achieve token-specific guidance using DG by selectively disrupting MAs in the target regions. In the revised draft, we have incorporated this token-specific guidance using Local DG into the Experiments section.
>
>
> ### **Unified view of CFG, Autoguidance, and DG**
> To facilitate a clearer conceptual understanding of DG and its relationship to existing guidance strategies, we present a unified formulation of CFG, Autoguidance, and DG in the following table.
>
> |Type|Formulation|Disruption|
> |:--:|:--:|:---:|
> |**CFG**|  $D_{\theta}\left(z_t, t,c\right)+w\left(D_{\theta}\left(z_t, t, c\right)-D_{\theta}\left(z_t, t,\hat{c}\right)\right)$|$c$|
> |**Autoguidance [1]**|  $D_{\theta}\left(z_t, t,c\right)+w\left(D_{\theta}\left(z_t, t, c\right)-D_{{\theta}^*}\left(z_t, t,c\right)\right)$|$\theta$|
> |**DG**| $D_{\theta}\left(z_t, t,c\right)+w\left(D_{\theta}\left(z_t, t, c\right)-D_{\theta}\left(\hat{z}_t, t,c\right)\right)$|$z_t$|
>
> *where **$\hat{c}$** denotes the unconditional prompt, **${\theta}^\*$** denotes under-capability model, and **$\hat{z}_t$** denotes $z_t$ with MAs-disrupted hidden state.*
>
> Under this unified formulation, the distinction becomes clear:
> - **CFG** constructs a degraded version by **"disrupting" the input prompt condition $c$**, resulting in **prompt-alignment guidance**.
> - **Autoguidance** constructs a degraded version by **"disrupting" the model $\theta$**, resulting in **entangled prompt-alignment and visual detail guidance**.
> - **Our DG** constructs a degraded version by **"disrupting" the visual input $z_t$ itself** through MA disruption, resulting in **visual detail guidance**.
>
> Our DG can be seamlessly combined with CFG, enabling a decoupled guidance mechanism where CFG handles prompt alignment while DG focuses on visual detail guidance. In the revised draft, we have added a new appendix section (Appendix K) to discuss the relationships and distinctions among CFG, autoguidance and DG.
>
> [1] Guiding a Diffusion Model with a Bad Version of Itself, NeurIPS 2024.

---

> ### Comment · Reviewer_SiyV · 2025-11-21
>
> I thank the authors for adding supporting experiments for local editing. I raise my rating to 6.

---

> > ### Author Response · Authors · 2025-11-22
> >
> > Thank you for your thoughtful review and for raising your rating. We sincerely appreciate your constructive feedback and are glad that the additional experiments addressed your concerns.

---

### Official Review · Reviewer_RLWh · 2025-10-29

**Soundness:** 3
**Presentation:** 4
**Contribution:** 3
**Rating:** 8
**Confidence:** 4

**Summary:**

This paper systematically studies the massive activations in the DiT (Diffusion Transformer) image generation framework. The authors find that these activations are highly correlated with detail synthesis. Based on this observation, they propose a guidance (DG) strategy to steer the generation process toward better texture synthesis. Experiments demonstrate its effectiveness.

**Strengths:**

1. The presentation is well-organized and easy to follow.
2. The study on massive activations is detailed and systematic.
3. The proposed DG strategy is effective and novel.

**Weaknesses:**

1. The paper lacks comparison with advanced CFG strategies, such as PAG [1], FA-CFG [2], and Semantic-CFG [3]. While such comparisons are not essential, including them would strengthen the experimental validation and enhance the robustness of the evaluation if avaialbe.

2. In Figure 2, the authors claim that massive activations appear at fixed dimensions across all patch tokens. Is this dimension consistent across all layers?

[1] Self-Rectifying Diffusion Sampling with Perturbed-Attention Guidance.
[2] FreCaS: Efficient Higher-Resolution Image Generation via Frequency-aware Cascaded Sampling.
[3] Rethinking the Spatial Inconsistency in Classifier-Free Diffusion Guidance.

**Questions:**

See weakness.

---

> ### Author Response · Authors · 2025-11-21
> **Response to Reviewer RLWh (1/2)**
>
> Thanks for your suggestions and for recognizing the novelty and contribution of our work. Please see the responses to your comments. In the revised draft, we mark our major revisions as “blue”.
>
> ### **Q1. Comparison with additional advanced CFG strategies.**
>
> Thank you for the valuable suggestions. We would like to kindly clarify that our DG has already been compared with several advanced CFG variants (e.g., CFG++ [4], CFG-Zero [5]) in Table 2 of the main paper, where DG demonstrates strong effectiveness in enhancing visual details. Following your suggestions, we additionally evaluate our DG with other advanced CFG strategies, including PAG [1], FA-CFG [2], and Semantic-CFG [3]. We report the results in *Table i*.
> - **DG versus PAG.** PAG constructs the "bad" version by modifying the self-attention maps of the conditional branch and does not require training an unconditional branch. As shown in *Table i*, DG achieves better performance than PAG (e.g., **30.14 vs. 29.20** on HPSv2.1 and **6.14 vs. 6.10** on Aesthetic), demonstrating superior effectiveness of our DG in enhancing visual details.
> - **DG versus advanced CFG.** We further compare our DG with advanced CFG strategies such as FA-CFG, Semantic-CFG, and CFG-Zero. DG requires no unconditional-branch training, yet still obtains the highest Aesthetic score (**6.14 vs. 6.07**) among all methods. When combined with CFG, DG achieves the best results on both HPSv2.1 (**30.96 vs. 30.57**) and Aesthetic score (**6.13 vs. 6.07**), underscoring its effectiveness and generality in improving visual detail quality.
>
>
> |Training uncond| Method |Anime|Concept| Painting |Photo|Avg.|Aesthetic|
> | :---: | --- | :---: |:---: |:---: |:---: |:---: |:---: |
> $\checkmark$|CFG |31.34|30.62|30.98|28.01|30.24 | 5.93
> |$\checkmark$|APG |30.76|29.98|30.24|26.86|29.46 | 5.89
> |$\checkmark$|CFG++ |31.58|30.32|30.95|27.24|30.02| 5.91
> |$\checkmark$|Semantic-CFG |30.92|29.99|30.92|29.16|30.25| 5.89
> |$\checkmark$|FA-CFG |31.07| 30.10|31.09|28.76|30.26| 5.96
> |$\checkmark$|CFG-Zero |31.64| 31.05|**31.35**|28.25|30.57| 6.07
> |$\times$|PAG |30.59| 28.92|29.38|27.91|29.20| 6.10
> |$\times$|DG (Ours)|31.14|30.17|30.05|28.70|30.14|**6.14**
> |$\checkmark$ |CFG+DG (Ours)|**32.23**|**31.11**|31.27|**29.21**|**30.96**| 6.13
>
> *Table i. Comparison with advanced CFG strategies on dataset HPSv2.1 with SD3. Training uncond: whether need to train an unconditional branch.*
>
> These results comprehensively verify the effectiveness of our DG in enhancing visual details. In the revised draft, we have incorporated these additional experiments and analyses into the Experiments section. Moreover, we have included a discussion of these advanced CFG strategies, including PAG, FA-CFG, and Semantic-CFG, in the Related Work section.
>
>
> ### **Q2. Are Massive Activations Dimensions consistent across all layers?**
>
> Yes, the dimensions of Massive Activations (MAs) are consistent across all layers for DiTs. We provide the additional visualizations of MAs dimensions across different layers in Figure 14 of Appendix. The results show a clear pattern: the dominant MAs dimensions remain consistent across all DiT layers. For example, in SD3, Massive Activations consistently occur at dimension 810 across all layers, and in SD3.5, the corresponding MA dimension remains fixed at 676.
>
> In the revised draft, we have revised the caption of Figure 2 and incorporated these additional experiments and analyses into the Appendix B section. The revised caption of Figure 2 is presented as follows:
> > **Massive Activations in DiTs.** The activation magnitudes of internal hidden states from the middle block ($k = N/2$) and timestep ($t=T/2$). We present the average magnitudes over 1,000 text prompts. Massive Activations (MAs) are consistently concentrated in a few fixed dimensions across all image patch tokens. The MA dimensions remain consistent across all layers (see Figure 14).
>
> [1] Self-Rectifying Diffusion Sampling with Perturbed-Attention Guidance, ECCV 2024.
>
> [2] FreCaS: Efficient Higher-Resolution Image Generation via Frequency-aware Cascaded Sampling, ICLR 2025.
>
> [3] Rethinking the Spatial Inconsistency in Classifier-Free Diffusion Guidance, CVPR 2024.
>
> [4] CFG++: Manifold-constrained Classifier Free Guidance for Diffusion Models, ICLR 2025.
>
> [5] CFG-Zero*: Improved Classifier-Free Guidance for Flow Matching Models, Arxiv 2025.

---

> ### Author Response · Authors · 2025-11-21
> **Response to Reviewer RLWh (2/2)**
>
> ### **Unified view of CFG, Autoguidance, and DG**
> To facilitate a clearer conceptual understanding of DG and its relationship to existing guidance strategies, we present a unified formulation of CFG, Autoguidance, and DG in the following table.
>
> |Type|Formulation|Disruption|
> |--|:--|:---:|
> |**CFG**|  $D_{\theta}\left(z_t, t,c\right)+w\left(D_{\theta}\left(z_t, t, c\right)-D_{\theta}\left(z_t, t,\hat{c}\right)\right)$|$c$|
> |**Autoguidance [1]**|  $D_{\theta}\left(z_t, t,c\right)+w\left(D_{\theta}\left(z_t, t, c\right)-D_{{\theta}^*}\left(z_t, t,c\right)\right)$|$\theta$|
> |**DG**| $D_{\theta}\left(z_t, t,c\right)+w\left(D_{\theta}\left(z_t, t, c\right)-D_{\theta}\left(\hat{z}_t, t,c\right)\right)$|$z_t$|
>
> *where **$\hat{c}$** denotes the unconditional prompt, **${\theta}^\*$** denotes under-capability model, and **$\hat{z}_t$** denotes $z_t$ with MAs-disrupted hidden state.*
>
> Under this unified formulation, the distinction becomes clear:
> - **CFG** constructs a degraded version by **"disrupting" the input prompt condition $c$**, resulting in **prompt-alignment guidance**.
> - **Autoguidance** constructs a degraded version by **"disrupting" the model $\theta$**, resulting in **entangled prompt-alignment and visual detail guidance**.
> - **Our DG** constructs a degraded version by **"disrupting" the visual input $z_t$ itself** through MA disruption, resulting in **visual detail guidance**.
>
> Our DG can be seamlessly combined with CFG, enabling a decoupled guidance mechanism where CFG handles prompt alignment while DG focuses on visual detail guidance. In the revised draft, we have added a new appendix section (Appendix K) to discuss the relationships and distinctions among CFG, autoguidance and DG.
>
> [1] Guiding a Diffusion Model with a Bad Version of Itself, NeurIPS 2024.

---

> ### Comment · Reviewer_RLWh · 2025-11-21
>
> We have no further concerns and decide to maintain the score as 8.

---

> > ### Author Response · Authors · 2025-11-21
> >
> > Thank you for your thoughtful review and for maintaining your positive assessment of our work. We appreciate your constructive feedback and are grateful for your support.

---

### Official Review · Reviewer_sJzY · 2025-10-31

**Soundness:** 3
**Presentation:** 3
**Contribution:** 3
**Rating:** 6
**Confidence:** 4

**Summary:**

This paper systematically studies the role of Massive Activations in image generation and discovers that they play a crucial role in generating fine-grained local details. Building on this observation, the authors propose a simple, training-free method that constructs a “detail-deficient” model by deliberately disrupting these Massive Activations. This degraded model is then used as a negative reference to guide the original network (similar to CFG) to generating images with better detail fidelity.

**Strengths:**

- The paper is well-motivated and well-structured. It first provide a clear analysis of Massive Activations in Diffusion Transformers, and then proposes a simple and effective method based on this observation to improve image generation quality.

- The proposed approach is simple and training-free. It only involves disrupting the massive activations within a pretrained model to construct a degraded variant, which is then used as a negative reference through a CFG-style guidance to enhance fine-grained visual details.

- Strong results. The proposed method shows strong performance gains and visual improvements

**Weaknesses:**

- DG relies on fixed-dimension activation patterns and the AdaLN scaling mechanism specific to DiTs. It is unclear whether the approach generalizes to non-transformer or hybrid architectures. Moreover, several recent works have suggested that AdaLN may not be the most optimal solution due to its parameter overhead and have proposed lighter alternatives. It would be valuable to discuss whether DG can be adapted to other schemes.

- No ablation on computational overhead or inference latency. Since the proposed method requires constructing and utilizing a degraded model, it would be helpful to provide a comparison of GPU memory usage and inference time with and without DG to better understand its efficiency trade-offs.


- It is good to also discuss the failure cases and limitations for the method. For example, while the finding that disrupting MAs mainly affects local details rather than semantics is compelling, such disruptions could also introduce unintended side effects. For exmaple I am gussing over-sharpening or loss of texture consistency. Additional experiments or qualitative examples discussing limitations would make the paper more comprehensive.

**Questions:**

- Can it be adapted to non-DiT or hybrid architectures?
- What is the computational overhead and inference latency of DG?

---

> ### Author Response · Authors · 2025-11-21
> **Response to Reviewer sJzY (1/3)**
>
> Thanks for your valuable suggestions. We’re glad that you found our work insightful and interesting. Please see the responses to your comments. In the revised draft, we mark our major revisions as “blue”
>
> ### **Q1. Adapting DG to other schemes.**
> Thank you for your insightful comments. We would like to demonstrate the generality of our DG method from two perspectives.
>
> **(1) Adapting DG to other architectures**
>
> Our DG strategy is a detail guidance approach driven by massive activations, and it is applicable to a wide range of models that exhibit massive activations, **not limited to the standard DiTs (e.g. SD3 and Flux)**. To assess the generality of our approach across different architectures, we evaluate our strategy on several other models, including efficient DiT and non-DiT models.
> - **Efficient DiT.** To examine the robustness of DG on efficient DiT, we conduct experiments with the popular PixArt-alpha [1]. PixArt-alpha adopts an efficient DiT architecture that injects text conditioning through **cross-attention** and employs **shared AdaLN layers** across all blocks to encode the timestep. These experiments allow us to specifically **verify whether DG remains applicable under the shared-AdaLN setting**.
> - **Non-DiT.** To further examine DG beyond transformer-based diffusion architectures, we conduct experiments with the recent SANA [2], which adopts a novel Linear-DiT design. SANA replaces all **standard self-attention with linear attention** and **incorporates 3×3 convolutions**, forming a **convolution-augmented hybrid architecture**. These experiments allow us to **verify whether DG remains effective under such a hybrid design** that incorporates convolutional operations.
>
> As shown in *Table i*, our DG strategy consistently improves the performance of both PixArt-alpha and SANA. In the conditional generation setting, DG yields substantial improvements in detail quality, **HPSv2.1 increases from 24.40 to 28.72 and the Aesthetic score from 5.91 to 6.12 for the Linear-DiT SANA model**. Moreover, when combined with CFG, DG further enhances visual quality, **improving HPSv2.1 from 30.13 to 30.52 and the Aesthetic score from 6.00 to 6.07**. These findings demonstrate that DG generalizes effectively beyond standard DiT architectures, delivering robust improvements in detail quality across both efficient DiT variants and non-DiT models. In the revised draft, we have added a new appendix section (Appendix F) to illustrate the generality of DG across different architectures and schemes.
>
>
> |Model|Type| DG |Clipscore|Blipscore | HPSv2.1 | Aesthetic|
> | ----- |:-----: | :-----: | :-------: |:-------: |:-------: |:-------: |
> PixArt-alpha|Cond| $\times$ | 22.64 | 68.41 | 25.63 |  6.01
>  ||Cond| $\checkmark$ | **23.43** | **72.07** | **29.18** | **6.53**
> ||CFG | $\times$ | **26.20** | **87.64** | 29.99 | 6.21
>  ||CFG| $\checkmark$ | 26.17 |86.88  | **30.74** | **6.34**
>  SANA|Cond| $\times$ | 23.52 | 78.25 | 24.40 |  5.91
>  ||Cond| $\checkmark$ | **24.98** | **84.11** | **28.72** | **6.12**
>  ||CFG | $\times$ | 27.07 | **91.03** | 30.13 | 6.00
>  ||CFG| $\checkmark$ | **27.20** | 90.25 | **30.52** | **6.07**
>
>
> *Table i. Evaluation of DG on Pixart-alpha and SANA under Conditional (Cond) and CFG settings. Clipscore and Blipscore for evaluating prompt alignment. HPSv2.1 and Aesthetic for evaluating detail quality.*
>
> **(2) Adapting DG for token-specific guidance**
>
> **We can achieve token-specific guidance using DG by selectively disrupting MAs in target tokens.** Specifically, we design a Local DG strategy in which we mask MAs exclusively for a selected subset of spatial tokens (e.g., tokens corresponding to a specific object or region), rather than masking MAs across all spatial tokens as in the original DG setting (see Figure 9).
>
> As demonstrated in Figure 9, Local DG effectively guides local details in the generated images. Specifically, by disrupting the MAs only in the tokens corresponding to the dog, **Local DG successfully enhances the local details of the dog while leaving other regions, such as the cat and background, essentially unaffected**. By exploiting this property, we can achieve token-specific guidance using DG by selectively disrupting MAs in the target regions. In the revised draft, we have incorporated this token-specific guidance using Local DG into the Experiments section.
>
> [1] PixArt-α: Fast Training of Diffusion Transformer for Photorealistic Text-to-Image Synthesis, ICLR 2024.
>
> [2] Sana: Efficient High-Resolution Image Synthesis with Linear Diffusion Transformer, ICLR 2025.
>
> [3] Guiding a Diffusion Model with a Bad Version of Itself, NeurIPS 2024.

---

> ### Author Response · Authors · 2025-11-21
> **Response to Reviewer sJzY (2/3)**
>
> ### **Q2. Computational overhead and inference latency**
>
> Thank you for the suggestion, we have added the computational overhead of our DG strategy in Appendix G. Specifically, we generate 100 images at 1024×1024 resolution using a single L40S (48GB) GPU. The average computational cost is reported in *Table ii*.
>
> **DG is approximately 1.5× faster than CFG and delivers superior performance.** As shown in *Table ii*, DG achieves higher performance than CFG (e.g., **6.16 vs. 6.01** on SD3.5 model). Moreover, DG generates a 1024×1024 image in 10.6s, **approximately 1.5× faster than CFG**, which requires 15.7s. This efficiency stems from DG’s architectural design: it leverages an MAs-disrupted conditional branch to guide the base model. Before the disruption depth (e.g., $m=20$ out of $N=38$ blocks for SD3.5), DG requires forwarding only the conditional branch, while CFG necessitates forwarding both the conditional and unconditional branches throughout the entire sampling process. This architectural difference makes DG both more efficient and effective than CFG.
>
> In the revised draft, we have added a new appendix section (Appendix G) to illustrate the computational overhead of DG.
>
> |Model| Type |GPU memory(GB)|Generation latency (s)| Aesthetic |
> | :---: | :---: | :---: |:---: |:---: |
> SD3 |Cond|17|2.3|5.58
> ||CFG|20|4.3|5.80
> ||DG (Ours)|20|3.5|6.01
> SD3.5 |Cond|28|7.2|5.94
> ||CFG|32|15.7|6.01
> ||DG (Ours)|32|10.6|6.16
> Flux |Cond|35|16.2|5.50
> ||CFG|42|36.0|5.96
> ||DG (Ours)|42|24.8|6.13
>
> *Table ii.  Computational overhead comparison for 1024x1024 image generation. Our DG method is approximately 1.5× faster than CFG and concurrently delivers superior performance.*
>
>
> ### **Q3. Failure cases and limitations**
> Thank you for your suggestions. We have added a new appendix section (Appendix L) to illustrate the failure cases of DG. The content of Appendix L is presented as follows:
> > In this section, we present visualizations and analyses of the failure cases associated with our DG strategy. DG is explicitly designed to enhance the local detail fidelity of generated images. However, this emphasis on fine-grained detail can occasionally compromise semantic faithfulness. In particular, when the prompt specifies strong stylistic, identity-related, or conceptual requirements, DG may favor detail enhancement over strict adherence to the intended semantics.
> >
> > As illustrated in Figure 22, DG produces outputs with noticeably richer textures and enhanced local details but may fail to fully satisfy the semantic requirements of the prompt. For instance, given the prompt $\textit{“The Mona Lisa as a vogue model, 1989 punk-inspired portrait, dramatic lighting, cinematic lighting”}$, DG generates a portrait with improved local details and textures, yet the output lacks the expected $\textit{vogue}$ style and the distinctive identity features of $\textit{Mona Lisa}$. A promising direction to address this limitation is to combine DG with CFG, enabling joint control over both local detail and semantic alignment. Moreover, when the guidance scale becomes relatively large (e.g., greater than 5), our DG strategy may also exhibit oversaturation effects, similar to those observed with CFG.
>
> Moreover, we have added a new appendix section (Appendix M) to discuss the limitations and future work of our paper. The newly added content is presented as follows:
> > **Limitations.** Our Detail Guidance (DG) method is primarily designed to explicitly enhance the detail quality of generated images. While it does improve prompt alignment, its ability to ensure strong alignment throughout the generation process is somewhat constrained. For prompts that specify strong stylistic, identity-related, or conceptual requirements, we recommend combining DG with CFG to provide joint guidance on prompt alignment and detail enhancement.
> >
> > **Future Work.** This paper mainly explores the potential of Massive Activations (MAs) during the sampling stage of DiTs. A more promising direction for future work lies in leveraging the capacity of MAs during the training stage. Since MAs play a crucial role in local detail synthesis, incorporating them more effectively into the training process could significantly enhance the performance of DiTs. This approach offers valuable insights for optimizing future DiTs, driving further advancements in their generative capabilities.

---

> ### Author Response · Authors · 2025-11-21
> **Response to Reviewer sJzY (3/3)**
>
> ### **Unified view of CFG, Autoguidance, and DG**
> To facilitate a clearer conceptual understanding of DG and its relationship to existing guidance strategies, we present a unified formulation of CFG, Autoguidance, and DG in the following table.
>
> |Type|Formulation|Disruption|
> |--|:--|:---:|
> |**CFG**|  $D_{\theta}\left(z_t, t,c\right)+w\left(D_{\theta}\left(z_t, t, c\right)-D_{\theta}\left(z_t, t,\hat{c}\right)\right)$|$c$|
> |**Autoguidance [1]**|  $D_{\theta}\left(z_t, t,c\right)+w\left(D_{\theta}\left(z_t, t, c\right)-D_{{\theta}^*}\left(z_t, t,c\right)\right)$|$\theta$|
> |**DG**| $D_{\theta}\left(z_t, t,c\right)+w\left(D_{\theta}\left(z_t, t, c\right)-D_{\theta}\left(\hat{z}_t, t,c\right)\right)$|$z_t$|
>
> *where **$\hat{c}$** denotes the unconditional prompt, **${\theta}^\*$** denotes under-capability model, and **$\hat{z}_t$** denotes $z_t$ with MAs-disrupted hidden state.*
>
> Under this unified formulation, the distinction becomes clear:
> - **CFG** constructs a degraded version by **"disrupting" the input prompt condition $c$**, resulting in **prompt-alignment guidance**.
> - **Autoguidance** constructs a degraded version by **"disrupting" the model $\theta$**, resulting in **entangled prompt-alignment and visual detail guidance**.
> - **Our DG** constructs a degraded version by **"disrupting" the visual input $z_t$ itself** through MA disruption, resulting in **visual detail guidance**.
>
> Our DG can be seamlessly combined with CFG, enabling a decoupled guidance mechanism where CFG handles prompt alignment while DG focuses on visual detail guidance. In the revised draft, we have added a new appendix section (Appendix K) to discuss the relationships and distinctions among CFG, autoguidance and DG.
>
> [1] Guiding a Diffusion Model with a Bad Version of Itself, NeurIPS 2024.

---

### Author Response · Authors · 2025-11-21
**Common Response**

Dear Reviewers and Area Chair,

We sincerely appreciate the time and effort you have dedicated to reviewing our manuscript.

In this work, we systematically investigate the role of Massive Activations (MAs) in Diffusion Transformers (DiTs) and discover that they play a crucial role in driving local detail synthesis. Building on this insight, we introduce Detail Guidance (DG), a simple yet effective self-guidance strategy that steers the model toward generating better visual details. As highlighted by the reviewers, our analysis of MAs is well-structured and provides meaningful new insights (all), and our proposed DG strategy is well-motivated (sJzY, RLWh), simple yet effective (all), demonstrated by strong empirical results and comprehensive experiments and analysis (all).

We greatly appreciate your constructive feedback. In response, we have carefully revised and strengthened the manuscript as follows:

- Token-specific guidance experiment using Local DG (Section 5.4)
- Comparison with additional advanced CFG methods (Table 2)
- Experiments demonstrating the generality of DG on other schemes (Appendix F)
- Computational overhead analysis of DG (Appendix G)
- Spatial map visualizations of MA dimensions (Appendix E)
- Visualization of Matched-non-MAs disruption (Figure 19)
- Detailed configurations for Figures 1–5 (Appendix H.1)
- A unified view of CFG, Autoguidance, and our DG (Appendix K)
- Failure-case analysis (Appendix L)
- Limitations and future work (Appendix M)

All modifications in the revised manuscript are highlighted in blue for your convenience to check.

We sincerely believe that these updates may further clarify the role of Massive Activations in DiTs and demonstrate the contributions and benefits of our proposed DG strategy to the ICLR community.


Thank you very much,

Authors.

---

### Comment · Area_Chair_sbUw · 2025-11-23
**The authors' rebuttal is available. Please read, comment, and discuss.**

Dear Reviewers,

Thanks for your time and effort in reviewing ICLR2026 submissions. The authors have provided their responses to your review. Please read and raise your further comments, and discuss with the authors.

Best regards,

Your AC

---

### Author Response · Authors · 2025-11-29
**Concise Summary**

Dear ACs, SACs and PCs,

We sincerely appreciate the time and effort you have dedicated to reviewing our manuscript. To support your evaluation, we would like to provide a concise summary for our paper.

### **1. Understanding the role of Massive Activations (MAs) in DiTs.**
Massive Activations (MAs) are widely present in Transformer architectures. In LLMs, MAs are closely associated with attention sink behavior and long-context learning [1][2]. In ViTs, they play a key role in global information processing [3]. In this work, we systematically investigate the role of MAs in Diffusion Transformers (DiTs) and reveal that they serve as a crucial mechanism driving local detail synthesis during generation.

### **2. Detail Guidance (DG): An MAs-driven self-guidance strategy for improving detail fidelity.**
Building on this insight, we introduce Detail Guidance (DG), a simple yet effective self-guidance strategy that steers the model toward generating better visual details.
Our DG can seamlessly integrate with Classifier-Free Guidance (CFG), enabling joint enhancement of detail fidelity and prompt alignment.

As highlighted by the reviewers, our analysis of MAs is well-structured and provides meaningful new insights (all), and our proposed DG strategy is well-motivated (sJzY, RLWh), simple yet effective (all), demonstrated by strong empirical results and comprehensive experiments and analysis (all).

We sincerely thank you again for your time, consideration, and the effort invested in the review process.

Best regards,

Authors


[1] Massive Activations in Large Language Models, COLM 2024.

[2] Efficient Streaming Language Models with Attention Sinks, ICLR 2024.

[3] Vision Transformers Need Registers, ICLR 2024.

---

### Meta-Review · Area_Chair_JsY5 · 2026-01-07

**Summary:**

This paper presents a systematic study of massive activations in Diffusion Transformer (DiT) models and shows that they are closely linked to the synthesis of fine-grained local details in generated images. By deliberately disrupting these activations, the authors construct a training-free, detail-deficient reference model and use it as negative guidance, analogous to classifier-free guidance, to steer the original model toward richer detail generation. The resulting Detail Guidance (DG) method enhances texture and local fidelity without retraining. Extensive analysis and experiments demonstrate that disrupting massive activations degrades local details, while DG consistently improves detail quality in image generation.

**Reviewer Concerns:**

Reviewers raised the following concerns:
1. Unclear generalizability to other architectures: DG relies on DiT-specific mechanisms (e.g., fixed-dimension activations and AdaLN), and it is unclear whether or how it can be generalized to other architectures or normalization schemes. (Reviewer sJzY)
2. Efficiency trade-offs are not analyzed: The paper does not report computational overhead, GPU memory usage, or inference latency for DG, even though it requires constructing and using a degraded model. (Reviewer sJzY)
3. Missing comparisons and stronger controls: Reviewers requested comparisons with advanced CFG strategies and related controls (e.g., under-trained or contrast models, blurred conditional-path controls), and noted that some existing controls (e.g., non-MA disruption) may not be matched in perturbation magnitude. (Reviewers RLWh and SiyV)
4. Method description needs clarification: It is unclear whether massive activations are measured or manipulated in the conditional or unconditional path, and key figure settings (e.g., layer, timestep, and whether MA dimensions are consistent across layers) are not sufficiently specified. (Reviewers RLWh and SiyV)
5. Limitations and failure cases are under-discussed: Reviewers requested more discussion or examples of failure modes and potential side effects (e.g., over-sharpening or reduced texture consistency). (Reviewer sJzY)

Overall, these concerns were addressed well in the rebuttal. Reviewers RLWh and SiyV acknowledged the rebuttal and kept or raised their scores to positive. The concerns from Reviewer sJzY (1, 2, 5) mainly requested additional analyses that were not essential for the core claims, and the authors added these analyses in the revised version.

**Reviewer Scores:**

The initial scores were 6, 8, and 4. After the rebuttal, Reviewer SiyV raised their score from 4 to 6, resulting in uniformly positive scores. The AC also agrees with the reviewers’ assessments and recommends acceptance.

---

### Decision · Program_Chairs · 2026-01-26

Accept (Poster)